# Indicators to evaluate the labor insertion of people with disabilities in conventional companies in Spain: A Delphi study

Rafael Sánchez-Arizcuren[1], Esperanza García-Uceda[2]\*, Bárbara Oliván-Blázquez[3]

1 Department of Physiatry and Nursing, University of Zaragoza, Spain, 2 Department of Business Organization, University of Zaragoza, Spain, 3 Department of Psychology and Sociology, University of Zaragoza, Spain

\* mariola@unizar.es

## Abstract

### Introduction

The level of labor integration of people with disability (PwD) is notably lower than that of people without disabilities. In order to evaluate the success of the labor market integration of people with disabilities, it is necessary to establish a series of indicators that go beyond hiring rates. Hence, the objective of this study is to develop a list of indicators with their specified individual weight that will serve to evaluate the success of the labor market insertion of PwD in conventional companies.

### Methodology

The Delphi method was used. In Phase 1, an open-ended questionnaire was distributed to 48 human resources and disability experts. From this qualitative analysis, a list of 52 indicators was drawn up and the experts were asked to evaluate their importance using a scale from 0 to 10 points. The convergence of opinion was obtained in the third phase. To evaluate the degree of stability of the experts' responses in the different phases, the Wilcoxon test was applied. Finally, a factor analysis was carried out in relation to the indicators.

### Results

After the development of the three phases and the factor analysis, a list of 26 indicators was obtained to measure the success of inclusive employability, structured in four factors: Work Performance, Labor Management, Social and Organizational Impact, Competency Assessment.

**Data availability statement:** The datasets used and/or analyzed during the current study are available in Zenodo Database (DOI https://doi.org/10.5281/zenodo.14757279). (DOI https://doi.org/10.5281/zenodo.14757280 in Spanish and https://doi.org/10.5281/zenodo.14858605 in english).

**Funding:** This work has been funded by the Department of Welfare and Family of the Government of Aragón (Spain) (OTRI Project: 2024/2009). The funders had no role in study design, data collection and analysis, decision to publish, or preparation of the manuscript.

**Competing interests:** The authors declare that the research was conducted in the absence of any commercial or financial relationships that could be construed as a potential conflict of interest.

## Conclusions

This project has provided a list of indicators that can be the basis for the creation of a scale to assess the successful integration of people with disabilities in the conventional company.

---

## Introduction

The employment of people with disabilities (PwD) in conventional companies presents both significant challenges and opportunities for modern societies. However, measuring success in workplace inclusion remains a challenge, due to the lack of specific tools that assess both individual adaptations and organizational practices. This issue not only responds to an ethical and human rights imperative [1], but also offers specific benefits for businesses and the economy in general [2]. According to the International Report on Disability, around one billion people, or 15% of the world's population, live with some form of disability and many of them face significant barriers to access formal employment [3]. In this context, having specific indicators would facilitate the objective evaluation of inclusive policies and practices, allowing companies to measure their impact and improve their strategies.

The level of labor integration of PwD is notably lower than that of people without disabilities, with lower activity rates that decrease even more with advancing age and in small municipalities [4]. In Spain, people with disabilities represent 6.33% of the working-age population, but their activity rate is significantly lower, reaching 35.3% in 2023 [5]. This gap highlights the need for instruments to identify not only the existing barriers, but also the actions required to overcome them and evaluate their effectiveness. This reality has prompted governments to establish specific regulations to improve the labor market integration of people with disabilities. An example of this is the Strategy on the Rights of Persons with Disabilities 2021–2030 approved by the European Commission in March 2021, which includes various measures to improve labor market outcomes for this group of people [6]. In Spain, the White Paper on Employment and Disability proposes a legislative and public policy framework for the employment and right to work of people with disabilities [7].

Currently, most working-age people with disabilities find employment in Occupational Centers and Special Employment Centers. However, it would be desirable to increase hiring in conventional companies due to the benefits that this integration offers to workers, companies and society in general [2]. The development of specific indicators that go beyond hiring rates and are adapted to the characteristics of PwD is crucial to evaluate the success of their inclusion in the regular company as well as to encourage the implementation of inclusive practices [5,7].

Although hiring rates are important, it is also necessary to include indicators that broaden the analysis of labor market insertion, such as work performance, productivity, quality of work performed, effectiveness and efficiency, salary, stability, attitude, skills and competencies, ability to work in a team, contribution to the work environment, training provided by the company or organization, level of participation

in decisions about the tasks performed or professional development, among others [8,9,10,11]. Since there are currently no specific indicators to measure the level of integration of people with disabilities in the labor market in the conventional company, it is necessary to adapt them to the particular circumstances of people with disabilities.

Therefore, when evaluating these indicators in the context of workers with disabilities, certain factors need to be taken into account. The most important are personal, social, and occupational factors, as well as performance or social integration through employment.

Among the personal factors, we can find that variables such as age, gender, and the type and degree of disability play a relevant role. Several studies have shown that women and people in certain age groups face a double difficulty in accessing the labor market due to disability and gender or age [12]. Taking into account the type of disability, it has been found that workers with intellectual disabilities or mental health problems experience greater difficulties to integrate into the labor market [13,14,15].

The social factors to be evaluated include the training, skills or abilities that the worker brings to the company, as well as other aspects to be considered in their incorporation. Pre-employment academic education, training provided by the company to the worker to facilitate their insertion, and training for co-workers are variables to be taken into account when PwD try to find a paid job, as well as in the performance and satisfaction of workers and employers. Several studies have shown how academic education determines access to the labor market, with workers with disabilities and university degrees having the highest employability rates [16], while the lack of educational qualifications leads to low-skilled and poorly paid jobs [17].

The work factors that affect the employability of people with disabilities are all those that can be found in the work environment, i.e., the work sector, the job position, the adaptations made, the job content (intrinsic interest of the job, variety, learning opportunities or difficulty), the salary and working conditions (working hours, breaks, environmental conditions), among others. These factors can influence employee performance and satisfaction [18].

However, a study conducted by the Foundation of the Spanish Organization of Blind People [19] in collaboration with FUNDOSA, Manpower and the European Social Fund, which analyzed the perspectives of employers regarding the hiring of people with disabilities and their performance, concluded that, according to these employers, 70.5% of workers with disabilities have the same performance as workers without disabilities, 7.9% have a higher performance, and 9.5% have a lower performance. A relevant aspect to take into account when analyzing the performance of a worker with a disability is the adequate adaptation of the job position to the type of disability, concluding that performance is higher when there is an adequate adjustment of the job position, as happens with workers without disabilities [20,21]. This adaptation is essential for the employment success of people with disabilities and their employers [22,23].

In this area, it is important to assess performance from a group perspective, i.e., it is not only relevant to evaluate the performance of the person with a disability in their professional tasks, but also to assess the extent to which the integration of a person with a disability in a group modifies the performance of other individuals [10]. It should also be taken into account that performance is also related to variables such as satisfaction and motivation [24,25].

Regarding the social integration of PwD through the workplace, most companies (68%) consider that hiring people with disabilities favors their positive social integration, above reasons such as economic or tax advantages or compliance with current legislation [26,27].

Considering the indicators used to measure the success of the labor market insertion of workers and the factors to be taken into account in the employability of PwD, the objective of this study is to develop a list of indicators with their specified individual weight that serves to evaluate the success of the labor market insertion of PwD in conventional companies.

## Materials and methods

This study was conceived as an exploratory research, and was developed in three phases between July and November 2023. The Delphi method was chosen due to its recognized ability to structure group communication and encourage

iterative reflection among experts in solving complex problems, especially in areas with limited evidence [28,29]. This approach has been widely validated in disciplines such as health and social sciences thanks to its effectiveness in building consensus, identifying discrepancies, and designing robust indicator or policy frameworks [30,31].

According to Okoli & Pawlowski [32], the Delphi method is useful in research that deals with the identification and prioritization of important issues for strategic decision making. It makes it possible to obtain informed and systematic opinions from a group of experts, while preserving their anonymity and guaranteeing the independence of their contributions. Furthermore, Flostrand, Pitt & Bridson [33] emphasize that the Delphi method is an effective tool for investigating complex problems from a multidisciplinary perspective, facilitating the integration of different points of view into a structured framework.

Its iterative structure, together with controlled feedback and statistical aggregation of responses, encourages convergence of opinion and informed reflection, reducing the biases of traditional group dynamics [34,35,36] (Fig 1). In addition, its ability to guarantee anonymity in the responses is essential to avoid hierarchical or social relationships between participants, thus reinforcing the validity and objectivity of the results obtained [32,33].

Although there are other qualitative techniques, such as individual interviews or focus groups, they do not offer the same guarantees of anonymity or the level of structuring that characterizes the Delphi method. Its ability to explore complex problems and arrange solutions through multiple rounds of questionnaires makes it particularly suitable for research in novel or data-limited areas [33,37].

Therefore, this method not only ensures the integration of expertise, but also involves a rigorous and efficient methodology for addressing research questions, allowing the development of a robust indicator framework from the contributions of experts from different disciplines [38].

Elaborated by the authors and adapted from the flow chart by Pozo et al., 2007, p. 355.

The research team conducting this study consists of five researchers with interdisciplinary expertise: a specialist in labor inclusion and public policy with extensive experience in participatory methods such as the Delphi technique, a specialist in business management, a statistician with expertise in data analysis, a professional specializing in social inclusion projects, and a researcher with extensive practical experience working directly with people with disabilities.

## Sample recruitment/Phase 0

According to Martínez [39], experts are defined as individuals with extensive knowledge and proven experience in the field, whose predictions are highly reliable. These experts possess the ability to provide well-founded evaluations of specific problems and offer actionable recommendations. In this study, the selection criteria were explicitly defined to include professionals with: a) a minimum of five years of experience in the field of inclusion and disability, b) proven expertise demonstrated through publications, project participation, or policy development, c) representation from diverse sectors, such as academia, the third sector, and conventional companies, to ensure a multidimensional perspective.

The selection process was guided by established frameworks, including the experts´ relationship to the research problem, professional experience, personal attributes relevant to participation in Delphi studies, and domain-specific expertise [40]. To preserve objectivity and minimize bias, all participants contributed anonymously and independently throughout the process, with measures in place to prevent contact between panel members. This approach enhanced the reliability of the data collected and ensured unbiased contributions from each expert [41].

The choice of the ideal number of experts depends on several factors, such as ensuring that it is sufficient, avoiding excessive loss of experts during repeated analysis phases, accessibility to experts, manageable workload, and speed in delivering preliminary results [42]. There are different recommendations, such as that of Landeta [43], who suggests that the appropriate range is between 7 and 30 experts, or that of Boulkedid et al. [44], who establish that the median number of panel members is 17, with an interquartile range of 11 to 31 experts.

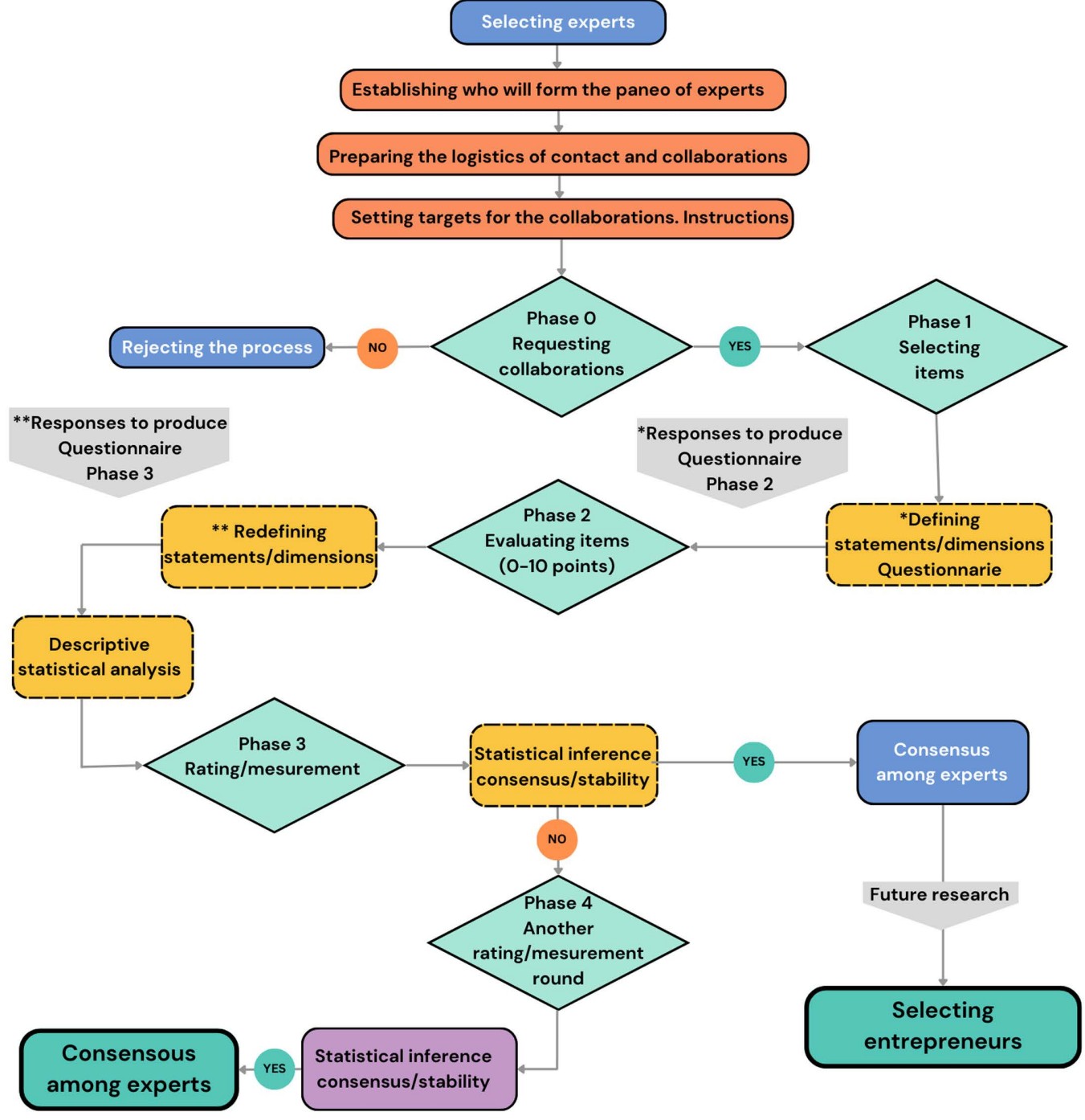

**Fig 1. Flow chart of the Delphi methodology.**

Initially (phase 0), 20 experts in disability and 16 experts in employability services for people with disabilities were contacted, as well as 38 representatives of conventional companies that hire people with disabilities. Of the 74 experts initially contacted, 48 agreed to participate in the study (response rate: 64,86%). This group was made up of 13 experts in the

field of disability, professionals from administrations and social entities, 16 experts in charge of employability services for people with disabilities offered by both social entities and foundations of conventional companies, and 19 experts belonging to conventional companies (employers hiring PwD).

The final selection of participants ensured compliance with the recommendations of Cabero & Barroso [42], that is, that the information was sufficient and manageable, that there was no excessive loss of experts during repeated rounds of analysis, that the experts were accessible, and that the speed in delivering preliminary results was adequate. Table 1 shows the characteristics of the sample of experts.

The experts were contacted by e-mail and provided with detailed information on the purpose of the research, the methodology employed, and the nature of their participation, which included the completion of at least three questionnaires. They were also asked to sign an informed consent document to formalize their collaboration.

## Phase 1

Phase 1 applied expert judgment to identify key indicators of inclusive employability. The first questionnaire was designed with open-ended questions to gather preliminary insights on labor inclusion for people with disabilities. Open-ended questions are particularly suitable for exploratory research as they allow for a broad and diverse range of initial perspectives to be captured before consolidating them into a structured questionnaire. This approach is consistent with the methodological recommendations for the Delphi technique, which emphasize the importance of qualitative input to establish a robust conceptual foundation [34,36,45].

The development of these questions followed a rigorous process, grounded in a comprehensive literature review on labor inclusion and human resource management. Key sources such as Peiró & Prieto [11], Schur et al. [46], and Lindsay et al. [47], guided the formulation of questions addressing critical aspects of recruitment and selection. For example, one question asked: "What criteria or aspects are important when evaluating the recruitment and selection process of people with disabilities in a conventional company? These questions were informed by studies that emphasize the significance of organizational policies and practices as pivotal areas for analysis.

Additionally, topics such as skills, job performance, and work environment were included in the questionnaire. These aspects were based on the frameworks proposed by Ishikawa [9] and Aguelo & Coma [8]. Their work highlights these dimensions as key indicators for assessing the success of people with disabilities in the workplace. By incorporating these themes, the questions aimed to gain a comprehensive understanding of the factors influencing the labor inclusion of people with disabilities.

In addition, the content of the questions on potential barriers and strategies for improvement was guided by recent research addressing specific challenges faced by people with disabilities (PwD) in the labor market, such as attitudinal barriers, accessibility issues, and support systems [14,47].

**Table 1. Description of the participating experts.**

| | | Frequency (percentage) |
|---|---|---|
| Gender | Male | 18 (37,5%) |
| | Female | 30 (62,5%) |
| Type of company | Conventional Service/Industry Companies | 19 (39,6%) |
| | Administrations and Foundations of Conventional Companies | 7 (14,6%) |
| | Third Sector Entities | 22 (45,8%) |
| Type of expert | Expert in Disability | 13 (27,1%) |
| | Employability Services Manager | 16 (33,3%) |
| | Employers/Human Resource Workers of Conventional Companies | 19 (39,6%) |

This review focused on areas such as recruitment practices, workplace culture, and specific employment barriers, which provided the basic framework for developing the open-ended questions.

An Expert Review was then carried out. A draft of the questions was carefully examined by a panel of five external experts. These experts, who possessed significant experience in disability inclusion, business management and qualitative research methods, provided critical feedback. This step ensured that the questions were relevant, clear, and aligned with the current challenges and trends in the field of labor inclusion.

Finally, an Iterative Refinement phase was carried out. Feedback from the experts´ review was incorporated to further refine the questions. This process ensured that the questions were open-ended yet sufficiently focused to elicit meaningful and actionable responses. This iterative approach enhanced the overall quality and accuracy of the questionnaire.

By starting with open-ended questions, the study ensured that the resulting indicators were based on diverse and comprehensive perspectives. This approach not only facilitated the identification of key themes but also provided the foundation for the subsequent phases of the Delphi process, where these initial inputs were refined and validated through thematic coding and consensus building.

Specifically, this first questionnaire consisted of seven open questions, as shown in Table 2, which were distributed to the 48 experts through an online platform.

A structured semantic analysis was used to interpret and categorize the answers obtained to these questions. This approach was designed to ensure the validity of the dimensions identified, following the recommendations of methodological studies on the Delphi method [29,48,49].

The responses were analyzed qualitatively following a systematic approach of categorization and synthesis [50,51]. This process involved several key steps to ensure the validity and relevance of the findings.

First, all open-ended responses were reviewed to identify key ideas and recurring concepts. Qualitative content analysis through thematic coding was used to structure the information [36]. Secondly, similar ideas were grouped into preliminary categories, such as recruitment and selection, job performance, and work environment. This grouping process was carried out by consensus among the researchers, ensuring rigor and objectivity in the interpretation of the responses [37].

Thirdly, the categories generated were evaluated for relevance, clarity and frequency among experts. Responses that were anecdotal, unclear, or not in line with the study objectives were systematically discarded to maintain methodological rigor [30,32]. Finally, the selected categories were transformed into specific items for the Phase 2 closed-ended questionnaire. Each item was carefully worded to ensure clarity and accuracy, capturing the essence of the insights provided by the experts [34,52].

An Expert Review was then conducted. A draft of the questions was carefully examined by a panel of five external experts with significant experience in disability inclusion and qualitative research methods, who provided critical feedback.

**Table 2. Open-ended questions asked to the experts in Phase 1.**

| |
| --- |
| 1. Which criteria or aspects are important when evaluating the recruitment and selection process of PwD in a conventional company? |
| 2. In your experience, what are the main relevant indicators or measures to evaluate the performance of PwD in a conventional company? |
| 3. List the main benefits that occur in the work environment when there is a person with a disability in the team in the conventional company. Are there any indicators or measures to evaluate them? Which ones? |
| 4. In your experience, can the fact of including a person with a disability in the team entail some kind of difficulty for that person and for the team? How could these difficulties be measured? |
| 5. What actions or strategies do you consider necessary to promote the inclusion and professional development of PwD in the conventional company? |
| 6. In your opinion, what are the main competences that should be worked on both in the person with disabilities hired and in the team in which they will be included in order to facilitate their adequate labor integration? |
| 7. Is there anything else you would like to add? or Is there any additional information relevant to this research? |

This step ensured that the questions were relevant, clear, and aligned with the current challenges and trends in the field of labor inclusion.

This structured process allowed for a robust translation of qualitative insights into measurable indicators, forming the basis for subsequent phases of the study. In addition, this approach ensures that the categories reflect both the diversity of views and alignment with dimensions recognized in previous research on labor inclusion. After this qualitative analysis, 52 indicators were obtained.

### Phase 2

Based on the theoretical dimensions obtained, a list of 52 indicators was drawn up and the experts were asked to evaluate the importance of each item using a scale of 0–10 points. In addition, in this second questionnaire, they were encouraged to propose improvements in the final wording of the items, as well as in the relevance and denomination of the dimensions into which they had been grouped. No suggestions were received to modify the wording or to incorporate additional items.

### Phase 3

After compiling the scores from the second round, a third questionnaire was designed with the aim of reaching a statistical consensus within the group of experts. Each participant was provided with two key pieces of information for each of the 52 indicators: (1) the average score assigned by the group in the second round; and (2) their own individual score from the same round. This design allowed participants to reflect on their position relative to the group consensus and to decide whether they wished to adjust their assessment. In cases where participants chose to modify their responses, they were asked to provide a rationale to capture the reasons for their adjustments.

However, no measures of dispersion, such as standard deviation, median, or interquartile range were shared with the participants. This decision intended to avoid cognitive overload and to keep the focus on the fit of individual responses relative to the group mean. Reducing statistical complexity minimized the risk of distraction, particularly for experts with limited experience in advanced statistical analysis. In addition, this approach sought to preserve the independence of the experts' judgment, reducing the influence that within-group variability might have on their assessments.

Although these measures of dispersion were not presented during the process, they were thoroughly analyzed and are included in the results section. This ensures that the variability and distribution of responses are fully considered in the analysis and interpretation of the data. This methodological decision reflects the balance between providing relevant information and maintaining the clarity and accessibility of the Delphi process [30].

To assess the possible convergence of opinion, the change in the responses received in the third phase with respect to the second phase was analyzed. We examined whether there had been variations in the scores given by the experts in the second phase once the group's mean ratings had been received. For this purpose, the "proportion of experts" statistic was used to verify that the average value of the responses in this third phase was within a range from [-0.5 to + 0.5], compared to the average value of the scores in the second phase. In the case of non-convergence of opinion, this methodology allows for as many rounds as necessary until convergence is achieved. After this convergence analysis, six items were statistically eliminated, leaving 46 indicators for the factor analysis.

To evaluate the degree of stability in the experts' responses in the different phases, a non-parametric test for related samples, the "Wilcoxon test" (H0 = No significant changes in the ratings), was used. This variable was the only one that did not follow a normal distribution. The objective was to identify the significance of the changes through the modification in the median, as well as their direction (positive or negative) and magnitude (mean value).

Finally, a factor analysis was performed on the answers given by the 48 experts in relation to the 46 indicators, applying the Principal Components method with Varimax rotation. Those indicators with communalities greater than 0.6 and with

loadings on a single factor were maintained in the structure of the list of indicators for the evaluation of the labor market insertion of PwD in conventional companies. Communalities were values ranging from 0 to 1. A communality of 1 indicates that the variable is strongly related to the extracted factor-dimension and all its variability is explained by it, while a communality of 0 indicates that none of the factors contribute to the variable. The magnitude of the factor loading indicates the strength of the relationship between the indicator and the factor-dimension: the higher the absolute value of the loading, the stronger the association. A loading close to 1 or -1 suggests a strong relationship, while a loading close to 0 indicates a weak relationship.

To complement and validate the results obtained in the main study, a follow-up survey was conducted among the experts who participated in the Delphi process. The objective of this survey was to evaluate the clarity of the process, the usefulness of the feedback received in each round, the impact of the justification of changes, the perception of the need for the third round and the applicability of the indicators developed. The survey consisted of 11 Likert-scale questions, where participants rated each item on a scale from 1 (extremely unclear, not useful, strongly disagree, etc.) to 10 (extremely clear, highly useful, strongly agree, etc.). Additionally, an open-ended section allowed experts to provide qualitative feedback. The survey was answered by 21 experts and the results were analyzed using descriptive statistics and qualitative analysis of open comments.

## Ethical issues

Ethical approval was granted by the Data Protection Unit of the University of Zaragoza (Spain) (Code 2023–179). The procedures carried out for the production of this work were adjusted to the ethical standards of the aforementioned unit and were in accordance with the 1975 Declaration of Helsinki. All subjects signed a written informed consent form; their data were anonymized and used only for research purposes.

## Results

### Phase 1

After sending the first questionnaire to the experts, which consisted of seven open questions, 1,031 contributions were received. These responses were analyzed individually, seeking consensus in the group of researchers to adequately interpret the information collected, which involved several rounds of analysis. As a result of this process, 50 indicators were initially obtained and grouped into five dimensions. These were: 1) Selection and Recruitment (SR), 2) Work Itinerary (WI), 3) Work Performance (WP), 4) Work Environment (WE), and 5) Socially Responsible Company (SRC). Subsequently, these indicators were formulated as quantitatively assessable items for use in Phases 2 and 3. In a second step within this phase, work was carried out under the premise that the resulting items should capture the conceptual essence of the experts' judgments. Therefore, a second content analysis was performed with the collaboration of five new experts working in different areas of specialization: a professional in organizational structures, an expert in recruitment and selection of human resources in conventional companies, a doctor of social work and two professional psychologists expert in human resources. These collaborators were encouraged to propose improvements both in the formulation of the items and in the definition of the dimensions.

On the one hand, their contribution highlighted the suitability of the proposed categorization of the indicators. On the other hand, they suggested changes in some statements and the incorporation of three new items: "to measure the change in mindset of the teams to work in an inclusive company" (SRC7. Mindset inclusive company), "to measure the change in mindset of the worker with disability in relation to their possibility of growth in the company" (IL7. Mindset PwD), and "to have an accessible complaints and claims system/protocol for the person with disability" (DT17. Accessible complaints and claims).

On the other hand, this panel of experts made a special point of not including as an item, "to know the average time to fill a position for a person with a disability (pwD)" (SR14. Average time to hire pwD), considering that inclusive

employability should be based on the principles of equality, non-discrimination and respect for privacy. Instead, they suggested promoting an inclusive work environment that values all candidates for their skills and competencies regardless of their disability.

Finally, the questionnaire included 52 indicators, which are shown in Table 3. It also shows the items added and eliminated by the five experts external to the panel.

## Phase 2

The list obtained after the first phase, which included 52 indicators, served as basis for the preparation of the second closed questionnaire corresponding to the second phase. In this new questionnaire, an open-ended question was added for additional comments on each item. The questionnaire was again distributed electronically to the 48 experts, asking them to evaluate the importance of each indicator on a scale of 0 to 10.

In addition, they were encouraged to provide suggestions to improve both the final wording of the items and the dimensions. However, no additional recommendations were received at this stage.

The second questionnaire was answered by 48 experts from different fields related to the labor inclusion of people with disabilities in ordinary companies. Each item included an open section for qualitative comments, where the experts could justify their score or make additional observations.

Regarding the qualitative contributions, it is worth mentioning that they provided a valuable context to interpret the results. Some relevant examples are highlighted below: Selection and Recruitment (SR 9): "It is essential to define the characteristics of the position in a clear and accessible manner to avoid initial barriers that may discourage candidates with disabilities."; Work Itinerary (WI2): "Training itineraries must be adapted not only to the type of disability, but also to the work context and the technological tools available."; Work Performance (WP10): "Problem solving is a competency that should be fostered in people with disabilities through specific development programs."; Work Environment (WE3): "Colleagueship can be significantly improved when awareness activities are conducted prior to the incorporation of a person with a disability."; Socially Responsible Company (SRC5): "The social and cultural benefits of including people with disabilities are difficult to quantify, but critical to strengthening organizational cohesion."

Table 4 shows the mean scores for each indicator. They were distributed as follows among the five categories: 13 indicators related to Selection and Recruitment (SR), 7 indicators related to the Work Itinerary (WI), 17 indicators related to Work Performance (WP), 8 indicators related to Work Environment (WE), and 7 indicators related to a Socially Responsible Company (SRC).

The table shows the mean, standard deviation (SD), minimum (min) and maximum (max) values, and percentiles 10 and 90. SR: Selection and Recruitment; WI: Work Itinerary; WP: Work Performance; WE: Work Environment; SRC: Socially Responsible Company; pWD: Person with disability; PwD: People with disability.

As can be seen, the distribution of the data is asymmetric and skewed to the right. This is inferred from the concentration of 39 mean scores above 8 and the absence of mean scores below 6.3. In addition, the 10th percentile is below 5 for only three questions, suggesting that most of the responses tend to be at the right end of the distribution. The variability in the data, as indicated by the standard deviation, ranges from 1.1 to 2.59. This corresponds to a higher concentration at the right end of the distribution, supporting the conclusion of an asymmetric distribution towards higher values. The high number of responses given by the experts corroborate the appropriate selection of questions, as only ten questions had a score of 0.

## Phase 3

To assess the possible convergence of opinion after sending the third questionnaire, we analyzed the change in the responses received in Phase 3 compared to Phase 2. Table 5 describes the convergence of responses in this third round, showing means and standard deviation. Supplementary S1 Table shows medians and interquartile ranges since the

**Table 3. List of indicators developed after Phase 1, and after consultation with a group of experts external to the initial panel.**

| DIMENSIONS | INDICATORS |
|---|---|
| SELECTION AND RECRUITMENT (SR) | SR1. To adapt the recruitment process when the candidate has a disability (simplifying the process, video calls with subtitles or sign language interpreters, accessible websites, etc.)<br>SR2. To adapt the selection process when the candidate has a disability (adjusting response times, providing clear job descriptions and ensuring accessibility in communication and testing).<br>SR3. To score the pwD on training.<br>SR4. To score the pwD on work experience.<br>SR5. To score the pwD on soft skills: assertive communication, teamwork, problem solving, etc.<br>SR6. To score the pwD on their job-specific technical competencies.<br>SR7. To score the pwD on their digital competencies.<br>SR8. To assess the external support (family/social/institutional) of the pwD.<br>SR9. To define job characteristics in a clear and accessible way.<br>SR10. To have people trained in disability in the selection process by the company.<br>SR11. To receive advice from external entities with specialists in the labor insertion of PwD.<br>SR12. To know the hiring rate of PwD in relation to inclusive recruitment and selection processes. SR13. To determine the percentage of candidates with disabilities in the selection process.<br>**SR14. To determine the average time to fill a position for a pwD. |
| WORK ITINERARY (WI) | WI1. To have an adapted incorporation protocol or welcome plan (employee training, welcome design) in the company.<br>WI2. To have adapted training and professional development itineraries for the pwD in the company (IT tools, communication and dissemination instruments, materials, specialized trainers).<br>WI3. To implement career plans for the pwD.<br>WI4. To have an internal mentor or tutor with specialized training in the company to accompany the pwD.<br>WI5. To measure the PwD retention rate.<br>WI6. To measure the promotion opportunities offered to the pwD.<br>*WI7. To measure the promotion of the "growth mindset" of the pwD. |
| WORK PERFORMANCE (WP) | WP1. To evaluate the work performance of the pwD according to general company parameters.<br>WP2. To measure the productivity of the work performed by the pwD.<br>WP3. To evaluate the performance taking into account the characteristics of the pwD.<br>WP4. To analyze workload in inclusive teams.<br>WP5. To analyze work rhythm in inclusive teams.<br>WP6. To measure the quality of the work performed by the pwD.<br>WP7. To assess the fulfillment of objectives by the pwD.<br>WP8. To assess the participation of the pwD in internal projects/programs.<br>WP9. To evaluate the participation of the pwD in meetings.<br>WP10. To assess the involvement of the pwD in problem solving.<br>WP11. To measure the pwD absenteeism.<br>WP12. To measure the punctuality of the pwD.<br>WP13. To assess the interaction and communication of the pwD with their colleagues.<br>WP14. To assess the empathy and assertiveness of the pwD.<br>WP15. To measure the level of autonomy of the pwD to perform the job tasks.<br>WP16. To assess the flexibility of the company in integrating the pwD (adapting punctuality, shifts, pace of work, etc.).<br>*WP17. To have a system/protocol for complaints and claims in the company that is accessible to the pwD. |
| WORK ENVIRONMENT (WE) | WE1. To assess the subjective perception of colleagues associated with the incorporation of a pwD.<br>WE2. To assess the identified disability training needs of the rest of the people in the company.<br>WE3. To assess the extent to which the incorporation of a pwD may affect the company's teamwork.<br>WE4. To assess the extent to which the incorporation of a pwD can improve the team's self-esteem.<br>WE5. To evaluate the extent to which the incorporation of a pwD can have an impact on innovation in teams.<br>WE6. To assess the team's ability to integrate and empathize with the pwD.<br>WE7. Importance of training the company's employees (senior management, middle management, colleagues) through awareness raising and education in order to promote the inclusion and professional development of the pwD.<br>WE8. To assess the perception of the pwD in relation to their feelings of acceptance and respect from their colleagues. |

*(Continued)*

**Table 3.** (Continued)

| DIMENSIONS | INDICATORS |
|---|---|
| SOCIALLY RESPONSIBLE COMPANY (SRC) | SRC1. To assess the importance of the benefits associated with the company's reputation when hiring PwD.<br>SRC2. To measure the accessibility of the environment.<br>SRC3. To measure the importance to employees of having an accessible environment (facilities, architectural barriers, workstations, etc.).<br>SRC4. To assess the benefits obtained by the pwD when performing their work.<br>SRC5. To value the benefits (economic, social, cultural) the company obtains by hiring PwD.<br>SRC6. To measure the financial cost of implementing adaptation and support measures and actions for the pwD.<br>*SRC7. To measure the change in mindset of the teams to work in an inclusive company. |

pwD: person with disability; PwD: people with disability.

* Indicators added by the group of five experts to the 50 indicators obtained by the research team in Phase 1.

** Indicator eliminated by the group of five experts from the 50 indicators obtained by the research team in Phase 1.

median was 0 in many items, due to the fact that more than 50% of the participants did not change their opinion. This analysis was carried out using several summaries: first, the summary with the proportion of experts who did not change their response with respect to the previous round. The mean value of the changes and their standard deviation show that the changes were minimal. For each question, the p-value of the Wilcoxon test is included, which corresponds to the null hypothesis that there is no change in the mean score.

Regarding changes in opinion between round 2 and round 3, 30% of the experts (14 out of 48) adjusted their responses on at least one item between both rounds. The changes were mainly minor, with average variations between -0.1 and +0.25 points.

The items with the highest frequency of change were: Selection and Recruitment (SR8): "Assess external support (family/social/institutional) of the person with disability" and Work Environment (WE5): "Impact of the incorporation of people with disabilities on team innovation". In the case of SR8, significant adjustments were recorded due to the experts´ interpretation of the importance of this indicator. For WE5, there was a debate about its relevance, which led some experts to modify their scores.

A marked convergence was found among the panel members, as 28 of the 48 experts did not make any changes to their initial ratings. Less than 5% of the participants made changes to their opinions, with a very slight average change. This change scored -0.1, 0.17, 0.21, 0.12 and 0.12 in the five categories respectively, indicating that the average change was in the range from -0.1 to +0.21. This leads to the conclusion that there were no substantial changes in the responses.

The table shows the number of experts that changed their mind (No. of changes), the proportion, mean, standard deviation (SD), minimum and maximum change value, and p-value of Wilcoxon test. SR: Selection and Recruitment; WI: Work Itinerary; WP: Work Performance; WE: Work Environment; SRC: Socially Responsible Company.

Furthermore, the result of the Wilcoxon test with a p-value > 0.05 for all indicators allows us to state that there were not significant changes in the responses between the two phases. Therefore, in the third phase no significant evolution was observed as a group, which determined that there would not be a fourth phase and consequently, the process of sending out questionnaires ended with this third mailing, as a high level of consensus had been reached among all participants. Therefore, it was considered that this degree of agreement would not increase with further consultations [45].

In a second step, we sought to identify the questions for which there were significant differences between the two groups of experts: those working in conventional companies that hire people with disabilities and the disability experts together with the employability service managers working in social entities of the third sector. The main objective was to compare the final scores between the two groups and to determine whether there were significant differences in their

responses. For this purpose, the medians were compared using the Wilcoxon test, as the data did not have a normal distribution.

The results showed that, for most of the questions in each block, the results were compatible in both groups. The Wilcoxon test was not significant at the 0.01 level in all questions related to Work Performance and Work Environment, while there were only two significant differences in Recruitment and Selection, and one in Socially Responsible Company. In Work Itinerary, three questions were found to have significant differences at the 0.01 level.

On the other hand, there were six indicators with significant discrepancies between the two groups of experts (p-value < 0.01). This shows that there was a notable discordance in the assessment between the group of conventional companies and the group of social entities of the third sector in the following indicators: "To adapt recruitment processes when the candidate has a disability" (SR1), "To have people trained in disability in the selection processes by the company" (SR10), "To Have IT tools, communication and dissemination instruments, materials, and specialized trainers for people with disabilities" (WI2), "To implement career plans for people with disabilities" (WI3), "To have an internal mentor or tutor with specialized training in the company to accompany the person with a disability" (WI4), and "To measure the importance for the workforce of having an accessible environment" (SRC3). This comparison and its p-values are shown in the supplementary material (S1 Table).

It is important to note that not all metrics are equally applicable in all organizational contexts. The approaches and objectives of the two groups analyzed differ: while professionals in third sector entities tend to focus on promoting inclusion, social equity, and creating a highly inclusive work environment with individualized attention to the needs of employees with disabilities, conventional companies prioritize operational efficiency, profitability, and productivity. Therefore, in order to achieve a more accurate assessment applicable to the objectives of each group and to improve the consistency and comparability of the data between the two groups, it was decided to eliminate these six indicators, leaving the final questionnaire with 46 items.

Once a consensus was reached, an analysis of the dimensions for measuring the success of inclusive employability was carried out by means of a factor analysis of the answers provided by the 48 experts in relation to the 46 indicators, applying the Principal Components method and Varimax rotation. Table 6 presents the results of the factor analysis, showing the indicators for measuring the success of inclusive employability, grouped into four factors or dimensions that explain 56.5% of the variability of the data. Finally, a list was obtained in which each factor groups the indicators from the highest to the lowest loading on the corresponding factor.

As can be seen, the five dimensions were transformed into four by identifying statistical patterns and validating the thematic coherence between the elements, which allowed for a more efficient and explanatory representation of the data. The indicator variables were more coherently integrated into the different dimensions identified by the analysis. Some variables were removed from the list of indicators because they had a very low correlation with the rest of the dataset, indicating that they did not contribute unique or substantial information to the overall pattern captured by the model. Others were discarded due to the very high correlation, interpreted as redundancy, and removed to simplify the model.

The percentage of explained variance was over 56%. Reliability indices were measured by Cronbach's Alpha, with an overall value of $\alpha = 0.90$, which is clearly higher than the reference value of 0.6 suggested by Peterson & Slater [53,54]. In addition, a KMO statistic of 0.75 was obtained, together with a significant Bartlett's test of sphericity, indicating that the factor analysis is appropriate for the database.

The results of the validation survey (Table 7) confirmed that most experts considered the Delphi process to be clear (M = 8.14, SD = 1.06) and that the feedback received was useful for reflecting on their responses (M = 7.95, SD = 1.43). However, the third round was not perceived as essential (M = 3.95, SD = 0.97), highlighting the need to discuss its impact on consensus generation. Regarding the applicability of the indicators in labor inclusion, the experts considered them useful (M = 7.86, SD = 1.19) and suitable for designing labor inclusion strategies (M = 8.19, SD = 1.40).

**Table 4. Description of the scores for each question in Phase 2 provided by the experts.**

### Selection and Recruitment (SR)

| Variable | Mean (SD) | Median (IQR) | min | Q10 | Q90 | max |
|---|---|---|---|---|---|---|
| SR1 Adapt recruitment process | 8.81 (1,65) | 9.5 (2) | 2 | 7 | 10 | 10 |
| SR2 Adapt selection process | 8.96 (1.3) | 9.5 (2) | 5 | 7 | 10 | 10 |
| SR3 Score the pwD on training | 7.46 (1.68) | 7 (3) | 4 | 5 | 10 | 10 |
| SR4 Score the pwD on work experience | 7.35 (1.93) | 7.5 (1.5) | 0 | 5 | 10 | 10 |
| SR5 Score the pwD on soft skills | 8 (1.52) | 8 (2) | 3 | 6.7 | 10 | 10 |
| SR6 Score the pwD on job-specific technical competencies. | 7.9 (1.59) | 8 (2) | 4 | 5.7 | 10 | 10 |
| SR7 Score the pwD on their digital competencies | 7.11 (1.63) | 7 (2) | 4 | 5 | 10 | 10 |
| SR8 Assess the external support | 6.54 (2.59) | 7 (4) | 0 | 3.7 | 10 | 10 |
| SR9 Define job characteristics | 9.44 (0.82) | 10 (1) | 7 | 8 | 10 | 10 |
| SR10 Have people trained in disability | 8.38 (1.62) | 9 (2) | 4 | 5.7 | 10 | 10 |
| SR11 Receive advice from external entities with specialists in labor insertion of PwD. | 8.69 (1.32) | 9 (2) | 5 | 7 | 10 | 10 |
| SR12 Hiring rate of PwD. | 6.64 (2.24) | 7 (3) | 0 | 4.7 | 9 | 10 |
| SR13% of candidates with disabilities in the selection process | 6.35 (2.33) | 7 (3) | 0 | 4 | 9 | 10 |

### Work Itinerary (WI)

| Variable | Mean (SD) | Median (IQR) | min | Q10 | Q90 | max |
|---|---|---|---|---|---|---|
| WI1 Adapted incorporation protocol/welcome plan | 9.12 (1.8) | 9 (1) | 5 | 8 | 10 | 10 |
| WI2 Adapted training and professional development itineraries | 8.81 (1.38) | 9 (2) | 5 | 7 | 10 | 10 |
| WI3 Implement career plans for the pwD | 7.71 (2.08) | 8 (2) | 0 | 5.7 | 10 | 10 |
| WI4 Specialized internal mentor to accompany the pwD | 8.4 (1.72) | 9 (2.25) | 3 | 6 | 10 | 10 |
| WI5. PwD retention rate. | 7.85 (1.79) | 8 (2) | 2 | 5 | 10 | 10 |
| WI6 Promotion opportunities for the pwD | 7.96 (1.82) | 8 (2.25) | 2 | 5.7 | 10 | 10 |
| WI7 Mindset of the pwD | 8.46 (1.37) | 8 (2) | 4 | 7 | 10 | 10 |

### Work Performance (WP)

| Variable | Mean (SD) | Median (IQR) | min | Q10 | Q90 | max |
|---|---|---|---|---|---|---|
| WP1 Work performance according to company parameters | 7.92 (1.87) | 8 (2) | 0 | 6 | 10 | 10 |
| WP2 PwD productivity | 8.04 (1.44) | 8 (2) | 5 | 6 | 10 | 10 |
| WP3 Performance of pwD | 8.23 (1.64) | 8 (3) | 3 | 6 | 10 | 10 |
| WP4 Workloads of inclusive teams. | 8.33 (1.88) | 9 (2.25) | 0 | 6.7 | 10 | 10 |
| WP5 Work rhythms in inclusive teams. | 8.35 (1.80) | 9 (2) | 0 | 6.7 | 10 | 10 |
| WP6 Quality of the work of the pwD. | 8.48 (1.25) | 9 (2.25) | 6 | 7 | 10 | 10 |
| WP7 Fulfillment of objectives by the pwD. | 8.81 (1.04) | 9 (2.) | 6 | 7.7 | 10 | 10 |
| WP8 Participation of the pwD in projects | 8.71 (1.03) | 9 (2) | 7 | 7 | 10 | 10 |
| WP9 Participation of the pwD in meetings. | 8.27 (1.85) | 8.5 (3) | 0 | 7 | 10 | 10 |
| WP10 Involvement of the pwD in problem solving | 8.52 (1.64) | 9 (2) | 0 | 7 | 10 | 10 |
| WP11 pwD absenteeism | 8.15 (1.60) | 8 (2) | 5 | 5.7 | 10 | 10 |
| WP12 pwD punctuality | 8.06 (1.86) | 8.5 (2) | 1 | 5.7 | 10 | 10 |
| WP13 pwD Interaction with colleagues | 8.29 (1.74) | 8.5 (2.25) | 1 | 7 | 10 | 10 |
| WP14 pwD empathy and assertiveness | 7.92 (1.83) | 8 (2) | 1 | 6 | 10 | 10 |
| WP15 Job task autonomy of the pwD. | 8.6 (1.12) | 8.5 (2) | 5 | 7 | 10 | 10 |
| WP16 Company flexibility in integrating the pwD | 8.79 (1.20) | 9 (2) | 5 | 7 | 10 | 10 |
| WP17 Accessible complaints and claims for the pwD | 8.9 (1.31) | 9 82) | 5 | 7.7 | 10 | 10 |

### Work Environment (WE)

| Variable | Mean (SD) | Median (IQR) | min | Q10 | Q90 | max |
|---|---|---|---|---|---|---|
| WE1 Perception of colleagues associated with the incorporation of a pwD | 7.83 (1.81) | 8 (2) | 2 | 6 | 10 | 10 |
| WE2 Training needs regarding disability in the company | 8.65 (1.21) | 9 (2) | 4 | 7 | 10 | 10 |

*(Continued)*

**Table 4.** (Continued)

**Selection and Recruitment (SR)**

| Variable | Mean (SD) | Median (IQR) | min | Q10 | Q90 | max |
|---|---|---|---|---|---|---|
| WE3 Impact on teamwork of the incorporation of a pwD | 8.21 (1.50) | 8.5 (2) | 5 | 6 | 10 | 10 |
| WE4 Improvement of the team's self-esteem due to the incorporation of a pwD | 8.21 (1.46) | 8 (2) | 5 | 6 | 10 | 10 |
| WE5 Impact of the incorporation of a pwD on innovation in teams D | 8.02 (1.72) | 8 (2) | 2 | 5.7 | 10 | 10 |
| WE6 Team's empathy with the pwD | 8.77 (1.10) | 9 (2) | 6 | 7 | 10 | 10 |
| WE7 Training of work team in disability | 8.96 (1.03) | 9 (2) | 6 | 8 | 10 | 10 |
| WE8 Perception of the pwD in relation to feelings of acceptance from colleagues | 8.94 (1.12) | 9 (2) | 6 | 7.7 | 10 | 10 |

**Socially Responsible Company (SRC)**

| Variable | Mean (SD) | Median (IQR) | min | Q10 | Q90 | max |
|---|---|---|---|---|---|---|
| SRC1 Reputation benefits when hiring PwD | 7.69 (1.94) | 8 (2) | 2 | 5 | 10 | 10 |
| SRC2 Accessibility of the environment | 8.83 (1.39) | 9 (2) | 5 | 7 | 10 | 10 |
| SRC3 Importance for the workforce of having an accessible environment | 8.79 (1.27) | 9 (2) | 5 | 7 | 10 | 10 |
| SRC4 Benefits obtained by the pwD when performing their work | 8.6 (1.35) | 9 (2) | 5 | 7 | 10 | 10 |
| SRC5 Benefits that the company obtains by hiring PwD | 8.31 (1.52) | 9 (1) | 3 | 6 | 10 | 10 |
| SRC6 Cost of adaptation and support measures and actions for the pwD. | 7.81 (1.73) | 8 (2) | 2 | 5.7 | 10 | 10 |
| SRC7 Change in mindset of the teams to work in an inclusive company. | 8.65 (1.06) | 9 (1.25) | 6 | 7 | 10 | 10 |

In addition to quantitative responses, some experts provided open-ended comments on the Delphi process. While several participants highlighted the value of the method in structuring the indicator assessment, others suggested that interased interaction among experts could have enriched the discussion and consensus-building process.

## Discussion

This study has developed, based on a Delphi methodology, a list of indicators to evaluate the success of work inclusion of people with disabilities in conventional companies, organized along the following dimensions: Work Performance (WP), Labor Management (LM), Social and Organizational Impact (S&OI), and Competency Assessment (CA). The results underline the importance of dealing with inclusion from multiple dimensions, ensuring a holistic approach that addresses both individual adaptations and organizational practices.

There are few studies that focus specifically on the evaluation of the employment success of people with disabilities in conventional companies. Most of the literature addresses general aspects of inclusion, such as barriers to enter the labor market and supportive public policies [55,56]. Research on the labor market inclusion of people with disabilities in conventional enterprises highlights both challenges and opportunities. Studies have found high rates of unemployment among this population, especially those with intellectual disabilities [57]. However, supported employment programs have shown a significant success in Spain [58,59]. While some companies implement inclusive practices as part of their corporate social responsibility [60], barriers to employment persist [61,62]. More research is needed to strengthen the business case for hiring PwD [63]. This study aims to contribute knowledge in this regard by developing a list of indicators to assess the employment success of people with disabilities in conventional companies. From this list, a scale could be developed to help measure and improve inclusive practices in conventional companies, creating more equitable and productive work environments.

The Delphi methodology made it possible to compile, categorize and analyze theoretical concepts on the inclusion of people with disabilities in conventional companies, weighing up the importance of each of them. The participation of different experts (third sector and conventional companies) has provided a broad perspective on the indicators of labor inclusion, allowing the development of a solid framework for this evaluation.

**Table 5. Description of the change in scores in Phase 3 with respect to Phase 2.**

**Selection and recruitment (SR)**

| Variable | Nº of changes | Proportion | mean | SD | min | max | p-value |
|---|---|---|---|---|---|---|---|
| SR1 Adapt recruitment process | 0 | 0 | 0 | 0 | 0 | 0 | -- |
| SR2 Adapt selection | 1 | 0.02 | 0.04 | 0.29 | 0 | 2 | 1 |
| SR3 Score the pwD on training | 1 | 0.02 | 0.04 | 0.29 | 0 | 2 | 1 |
| SR4 Score the pwD on work experience | 1 | 0.02 | -0.04 | 0.29 | -2 | 0 | 1 |
| SR5 Score the pwD on soft skills | 2 | 0.04 | 0.02 | 0.33 | -1 | 2 | 1 |
| SR6 Score the pwD on job-specific technical competencies. | 1 | 0.02 | -0.02 | 0.14 | -1 | 0 | 1 |
| SR7 Score the pwD on digital competencies | 2 | 0.04 | 0.07 | 0.36 | 0 | 2 | 0.371 |
| SR8 Assess the external support | 5 | 0.1 | -0.1 | 0.63 | -3 | 2 | 0.341 |
| SR9 Define job characteristics | 2 | 0.04 | 0.06 | 0.32 | 0 | 2 | 0.371 |
| SR10 Have people trained in disability | 1 | 0.02 | 0.04 | 0.29 | 0 | 2 | 1 |
| SR11 Receive advice from external specialists in labor insertion of PwD. | 0 | 0 | 0 | 0 | 0 | 0 | -- |
| SR12 Hiring rate of PwD | 4 | 0.08 | -0.07 | 0.74 | -3.5 | 3 | 0.584 |
| SR13% of candidates with disabilities in the selection process | 4 | 0.08 | -0.08 | 0.92 | -4 | 4 | 0.71 |

**Work Itinerary (WI)**

| Variable | Nº of changes | Proportion | mean | SD | min | max | p-value |
|---|---|---|---|---|---|---|---|
| WI1 Adapted welcome plan | 1 | 0.02 | 0.02 | 0.14 | 0 | 1 | 1 |
| WI2 Adapted training and professional development itineraries | 0 | 0 | 0 | 0 | 0 | 0 | -- |
| WI3 Implement career plans for the pwD | 3 | 0.06 | -0.12 | 0.53 | -3 | 0 | 0.181 |
| WI4 Specialized internal mentor that accompanies PwD | 1 | 0.02 | -0.06 | 0.43 | -3 | 0 | 1 |
| WI5. PwD retention rate. | 4 | 0.08 | 0.17 | 1.12 | -1 | 7 | 0.581 |
| WI6 Promotion opportunities for the pwD | 3 | 0.06 | 0.1 | 0.63 | -1 | 3 | 0.414 |
| WI7 Mindset of the PwD | 2 | 0.04 | 0.04 | 0.2 | 0 | 1 | 0.346 |

**Work Performance (WP)**

| Variable | Nº of changes | Proportion | mean | SD | min | max | p-value |
|---|---|---|---|---|---|---|---|
| WP1 Work performance according to company parameters | 2 | 0.04 | 0.06 | 0.32 | 0 | 2 | 0.371 |
| WP2 pwD productivity | 1 | 0.02 | 0.04 | 0.29 | 0 | 2 | 1 |
| WP3 Performance of the pwD | 0 | 0 | 0 | 0 | 0 | 0 | -- |
| WP4 Workloads of inclusive teams. | 2 | 0.04 | 0.15 | 0.77 | 0 | 5 | 0.371 |
| WP5 Work rhythms in inclusive teams. | 3 | 0.06 | 0.21 | 0.87 | 0 | 5 | 0.181 |
| WP6 Quality of the work of the pwD. | 2 | 0.04 | 0.06 | 0.32 | 0 | 2 | 0.371 |
| WP7 Fulfillment of objectives by the pwD. | 0 | 0 | 0 | 0 | 0 | 0 | -- |
| WP8 Participation of the pwD in projects | 1 | 0.02 | 0.02 | 0.14 | 0 | 1 | 1 |
| WP9 Participation of the pwD in meetings. | 1 | 0.02 | 0.02 | 0.14 | 0 | 1 | 1 |
| WP10 Involvement of the pwD in problem solving | 2 | 0.04 | 0.06 | 0.32 | 0 | 2 | 0.371 |
| WP11 pwD absenteeism | 3 | 0.06 | 0.04 | 0.35 | -1 | 2 | 0.586 |
| WP12 pwD punctuality | 1 | 0.02 | -0.02 | 0.14 | -1 | 0 | 1 |
| WP13 pwD interaction with colleagues | 1 | 0.02 | 0.02 | 0.14 | 0 | 1 | 1 |
| WP14 pwD empathy and assertiveness | 2 | 0.04 | 0.04 | 0.2 | 0 | 1 | 0.346 |
| WP15 Job task autonomy of the pwD. | 2 | 0.04 | 0 | 0.21 | -1 | 1 | 1 |
| WP16 Company flexibility for the incorporation of PwD | 1 | 0.02 | -0.02 | 0.14 | -1 | 0 | 1 |
| WP17 Accessible complaints and claims | 2 | 0.04 | -0.04 | 0.2 | -1 | 0 | 0.346 |

**Work Environment (WE)**

| Variable | Nº of changes | Proportion | mean | SD | min | max | p-value |
|---|---|---|---|---|---|---|---|
| WE1 Perception of colleagues associated with the incorporation of a pwD | 4 | 0.08 | 0.04 | 0.77 | -1 | 5 | 0.85 |
| WE2 Training needs regarding disability in the company | 2 | 0.04 | 0 | 0.21 | -1 | 1 | 1 |

*(Continued)*

**Table 5.** (Continued)

**Selection and recruitment (SR)**

| Variable | Nº of changes | Proportion | mean | SD | min | max | p-value |
|---|---|---|---|---|---|---|---|
| WE3 Impact on teamwork of the incorporation of a pwD | 3 | 0.06 | 0.04 | 0.62 | -1 | 4 | 1 |
| WE4 Improvement of the team's self-esteem due to the incorporation of a pwD | 3 | 0.06 | 0 | 0.36 | -1 | 2 | 1 |
| WE5 Impact on innovation in team due to the incorporation of a pwD | 5 | 0.1 | 0.12 | 0.7 | -1 | 3 | 0.276 |
| WE6 Team's empathy with the pwD | 2 | 0.04 | -0.04 | 0.2 | -1 | 0 | 0.346 |
| WE7 Training of work team in disability | 4 | 0.08 | 0.02 | 0.39 | -1 | 2 | 0.85 |
| WE8 Perception of the pwD in relation to feelings of acceptance from colleagues | 1 | 0.02 | -0.02 | 0.14 | -1 | 0 | 1 |

**Socially Responsible Company (SRC)**

| Variable | Nº of changes | Proportion | mean | SD | min | max | p-value |
|---|---|---|---|---|---|---|---|
| SRC1 Reputation benefits when incorporating PwD | 2 | 0.04 | 0 | 0.62 | -3 | 3 | 1 |
| SRC2 Accessibility of the environment | 2 | 0.04 | 0.04 | 0.2 | 0 | 1 | 0.346 |
| SRC3 Importance for the workforce of having an accessible environment | 1 | 0.02 | 0.04 | 0.29 | 0 | 2 | 1 |
| SRC4 Benefits obtained by the pwD for working | 1 | 0.02 | 0.12 | 0.87 | 0 | 6 | 1 |
| SRC5 Company benefits by incorporating PwD | 2 | 0.04 | 0.06 | 0.32 | 0 | 2 | 0.371 |
| SRC6 Cost of adaptation and support measures and actions for the pwD. | 1 | 0.02 | 0.1 | 0.72 | 0 | 5 | 1 |
| SRC7 Mindset in the teams to work in an inclusive company. | 0 | 0 | 0 | 0 | 0 | 0 | -- |

The results of this study are consistent with other research in key areas of employment inclusion of people with disabilities, which strengthens the validity of the findings and recommendations presented. The importance of adapting recruitment and selection processes to include people with disabilities is also highlighted by González-González et al. [64]. In addition, continuous training and awareness-raising for all employees is emphasized Ortíz-Marcos et al. [65] and Patras et al [66] as fundamental to overcome stereotypes and corporate rejection. The need to establish mentoring and ongoing support programs with specialized mentors is also reflected in the findings of Lindsay et al. [47]. The accessibility of the environment and adaptations in the workplace are also addressed in the ILO report [67]. Other studies [46] have also highlighted the importance of flexibility policies for effective inclusion.

The findings of this study reinforce the importance of assessing occupational performance and implementing reasonable accommodations using objective indicators, consistent with Schur et al. [46] and Lindsay et al. [47]. These authors emphasize that effective inclusion goes beyond hiring, requiring fair performance assessments and accommodations that enable full participation. In addition, reasonable accommodations can enhance productivity and work environment by reducing structural barriers. In this regard, the indicators developed in this study offer a valuable tool for measuring and improving inclusive employability, helping companies optimize their inclusion strategies and long-term sustainability. The use of dimensions such as organizational and social impact shows similar findings in research highlighting the benefits of diversity for work environment and innovation [64,68].

However, some results are not consistent with the literature consulted. The low importance given by the experts to indicators related to the average hiring time or the percentage of PwD in selection processes contrasts with studies that highlight these elements as fundamental for inclusion [60]. These discrepancies could be explained by differences in the priorities of the participating expert groups and suggest the need to explore more inclusive approaches.

In addition, the implementation of reasonable accommodations, along with continuous awareness-raising for all employees, stand as essential measures to overcome barriers and stereotypes. Research evidences that reasonable accommodations do not represent a significant burden for companies and instead, contribute to talent retention and improved work environment [69,70,71].

**Table 6. Factorial analysis. Variables for assessing success in the labor market integration of people with disabilities.**

| Factors/ Explained Variance/ α* | Indicators to evaluate the labor insertion of people with disabilities in conventional companies | Components | | | |
|---|---|---|---|---|---|
| **WP: Work Performance** 17,05% α*=0,912 | WP12 To measure the punctuality of the pwD | 0,853 | | | |
| | WP9. To evaluate the participation of the pwD in meetings | 0,806 | | | |
| | WP10. To assess the involvement of the pwD in problem solving | 0,799 | | | |
| | WP15. To measure the level of autonomy of the pwD to perform the job tasks | 0,702 | | | |
| | WP14. To assess empathy and assertiveness of the pwD | 0,695 | | | |
| | WP13. To assess the interaction and communication of the pwD with colleagues | 0,688 | | | |
| | WP16. To assess the flexibility of the company in integrating PwD | 0,645 | | | |
| | WP7. To assess the fulfillment of objectives by the pwD | 0,610 | | | |
| **LM: Labor Management** 14,20% α*=0.87 | WI6. To measure the promotion opportunities offered to the pwD | | 0,814 | | |
| | WP5. To analyze work rhythms in inclusive teams | | 0,727 | | |
| | WP4. To analyze workloads in inclusive teams | | 0,706 | | |
| | WI5. To measure the PwD retention rate | | 0,694 | | |
| | WP3. To evaluate the performance taking into account the characteristics of a pwD | | 0,618 | | |
| **S&OI: Social and Organizational Impact** 13,55% α*=0.82 | WE4. To assess the extent to which the incorporation of a pwD can affect the improvement of the team's self-esteem | | | 0,898 | |
| | WE6. To assess the team's ability to integrate and empathize with the pwD | | | 0,733 | |
| | WE5. To evaluate the extent to which the incorporation of a pwD can have an impact on innovation in teams | | | 0,694 | |
| | WE7. Importance of training the company's employees in order to promote the inclusion and professional development of the pwD | | | 0,621 | |
| | SRC5. To value the benefits (economic, social, cultural..) the company obtains by incorporating PwD | | | 0,618 | |
| | SR13. To determine the percentage of candidates with disabilities in the selection process | | | 0,615 | |
| | WE3. To assess the extent to which the incorporation of a pwD may affect the company´s teamwork | | | 0,612 | |
| **CA: Competency assessment** 11,67% α*=0.87 | SR7. To score the pwD on their digital competencies. | | | | 0,839 |
| | SR6. To score the pwD on their job-specific technical competencies | | | | 0,793 |
| | SR5. To score the pwD on soft skills | | | | 0,756 |
| | SR3. To score the pwD on training | | | | 0,726 |
| | SR4. To score the pwD on work experience | | | | 0,702 |
| | WP1. To evaluate the work performance of the pwD according to general company parameters | | | | 0,657 |

Total Variance: 56,5%. Residues:43,5%. Alfa: 0,90. KMO: 0,75.

The indicators developed in this study can be the basis for the creation of useful tools (which the research team is already working on) to evaluate and promote the inclusion of people with disabilities in mainstream organizations, to design internal inclusion policies that respond to the specific needs of each organization, to conduct accessibility audits and evaluate inclusive practices, as well as to facilitate the monitoring of the impact of inclusion initiatives in the workplace [72,73].

Sustainability of inclusive practices is essential to ensure that initiatives are not limited to the short term but transform organizational culture in a lasting way [74,75,76]. The indicators developed in this study seek to foster these structural and lasting changes. In addition to assessing current practices, they have the potential to drive sustainable organizational change that benefits both people with disabilities and overall business performance [77,78].

It is necessary to take into account that the implementation of the indicators requires a collaborative approach involving diverse stakeholders, both within the organization (such as disabled and non-disabled employees, human resource managers and team leaders) and outside (e.g., specialized consultants or independent auditors) [79,80,81,82].

This study has both strengths and limitations. One of the strengths is that it addresses a current and globally relevant issue: the inclusion of people with disabilities (PwD) in the labor market. This topic aligns with international efforts to promote inclusive workplaces and fills a critical gap by providing tools to assess and improve labor inclusion [46,67].

The use of the Delphi methodology is another key strength, as its iterative and consensus-based approach ensures an exhaustive collection of expert input, integrating diverse perspectives and adding credibility and depth to the findings [31,33]. The inclusion in the expert panel of employers and professionals in the field of human resources in conventional companies, disability experts, and managers of employability services, as well as people with disabilities working in the field of human resources within disability-focused organizations, reflects the integration of diverse perspectives and is also a strength of this study.

Although the results of the follow-up survey indicated that the third round of the Delphi process was not perceived as essential by some experts, they also validated the usefulness of the indicators generated. This suggests that, although methodological design can be improved in future studies, the results obtained are robust and relevant. Validation in real workplace settings is recognized as a crucial next step to strengthen the applicability of the indicators.

Qualitative comments from the experts reflect mixed perceptions of the Delphi process. While several participants emphasized the usefulness of the method in identifying key indicators, some suggested that more interaction among experts might have supported a more dynamic consensus. In future research, it would be advisable to promote more interaction among experts.

Finally, the statistical validation of the indicators through factor analysis increases the robustness of the study, providing a solid basis for their practical application in various organizational settings [83].

**Table 7. Results of the validation survey.**

| | Mean | Median | SD | IQR | Min | Max |
|---|---|---|---|---|---|---|
| How would you evaluate the Delphi process used in this study? | 8.14 | 8 | 1.062 | 1.00 | 6 | 10 |
| To what extent did the feedback received in each round help you to evaluate your responses? | 7.95 | 8 | 1.431 | 2.00 | 5 | 10 |
| In the second round, do you think the information provided (group average and your previous response) was sufficient to reflect on your answers? | 8.33 | 9 | 1.155 | 1.00 | 6 | 10 |
| Do you believe that the requirement to justify changes in your responses influenced your decision to modify or maintain your answers? | 7.19 | 7 | 1.806 | 3.00 | 3 | 10 |
| To what extent do you believe that the consensus reached reflects genuine and well-founded expert opinions? | 8.14 | 9 | 1.352 | 2.00 | 6 | 10 |
| Do you think the third round was necessary to consolidate consensus or improve the quality of the responses? | 3.95 | 4 | 0.973 | 2.00 | 2 | 5 |
| Do you consider that indicators resulting from this study applicable for evaluating labor inclusion in conventional companies? | 7.86 | 8 | 1.195 | 2.00 | 6 | 10 |
| Do you believe the results of this study are practically useful for designing labor inclusion strategies? | 8.19 | 8 | 1.401 | 2.00 | 5 | 10 |
| Would you recommend conducting future studies that include validation in real work environments before applying these indicators on a larger scale? | 8.10 | 9 | 1.670 | 2.00 | 4 | 10 |

On the other hand, limitations include the lack of a formal pilot test of the second questionnaire with an independent group of participants. However, the questionnaire was thoroughly reviewed by a group of five new independent experts, who assessed its clarity, relevance, and alignment with the study objectives.

Nevertheless, the scalability of the indicators and the implementation of a pilot study with a new group of experts are among the actions planned for the next phase of the project.

In addition, depending on the type of the worker´s disability, some indicators may not be appropriate and this should be taken into account when developing a measurement scale. Also, the applicability of these indicators may vary according to different organizational contexts and economic sectors, as each industry and type of company has unique characteristics. Another limitation is that some indicators may require specific adaptations to be applicable in certain work environments, such as smaller or less resourced enterprises. Future research could replicate this model in other geographical contexts and sectors to assess its validity and adaptability. Including a broader group of participants, both in terms of industries and regions, would strengthen the robustness and applicability of the indicators.

While the elimination of six indicators was intended to ensure the consistency and usefulness of the results, we recognize that their exclusion may have limited the opportunity to further explore differences of opinion between the expert groups. These divergent indicators fostered deeper discussion within the research team, which included a deliberation about the potential value of retaining these indicators and the potential risk of diluting the clarity and focus of the final results. However, this process underscored the importance of transparently documenting methodological decisions and highlighted the need for future research to analyze divergent opinions as part of a more comprehensive exploration of labor market inclusion challenges.

Another limitation is that the present study lacks an independent external evaluation of the quality of the development of the Delphi process. Although the study design included a carefully selected panel of experts and iterative validation of the indicators during the early phases, we recognize that the incorporation of additional review by external experts could have enriched the external validity and applicability of the results.

While the Delphi method used in this study effectively structured expert input, it did not facilitate direct interaction among participants, which may have limited opportunities for more in-depth discussion. Additionally, some experts perceived the third round as less essential for refining consensus. However, a follow-up validation survey reinforced the relevance of the indicators developed, confirming their perceived usefulness and applicability in assessing labor inclusión.

Another aspect is that, although the indicators were statistically validated, they need to be tested in diverse real-world work contexts and on a larger scale to assess their practical applicability and effectiveness.

Finally, a crucial aspect for the labor inclusion of people with disabilities is technological accessibility. We recognize its importance and have identified the need to develop specific indicators to assess the adoption and effectiveness of accessible technologies. For future research, it should be noted that the items related to this may be highly variable over a short period of time.

## Conclusions

This study has identified 26 key indicators for evaluating the labor inclusion of people with disabilities in conventional companies using the Delphi methodology. Further research is needed to analyze the suitability of these indicators in different contexts and types of disability. In addition, these indicators could serve as a reference for the creation of a scale to assess the labor inclusion of this group of people in conventional companies, thus providing tools that support labor inclusion.

## Supporting information

**Table S1. Comparison of the scores of the two types of experts in third phase.**
(DOCX)

## Acknowledgments

We would like to thank all the experts who contributed their knowledge during the different phases of the study, as well as the rest of the researchers involved in the study.

We also wish to thank to the University of Zaragoza, the Research Group CREVALOR (S42_17R), the Aragonese Primary Care Research Group (GAIAP, B21_23R) that is part of the Department of Innovation, Research and University in the Government of Aragón (Spain) and the Institute for Health Research Aragón (IIS Aragón); the Research Network on Chronicity, Primary Care and Health Promotion (RICAPPS, RD21/0016/0005) that is part of the Results-Oriented Cooperative Research Networks in Health (RICORS) (Carlos III Health Institute); and ERDF Funds "A way of making Europe".

## Author contributions

**Conceptualization:** Esperanza García-Uceda.

**Data curation:** Rafael Sánchez-Arizcuren, Esperanza García-Uceda.

**Formal analysis:** Bárbara Oliván-Blázquez.

**Funding acquisition:** Esperanza García-Uceda.

**Investigation:** Rafael Sánchez-Arizcuren, Esperanza García-Uceda, Bárbara Oliván-Blázquez.

**Methodology:** Rafael Sánchez-Arizcuren, Esperanza García-Uceda, Bárbara Oliván-Blázquez.

**Project administration:** Esperanza García-Uceda.

**Resources:** Esperanza García-Uceda.

**Validation:** Esperanza García-Uceda.

**Visualization:** Bárbara Oliván-Blázquez.

**Writing – original draft:** Rafael Sánchez-Arizcuren, Bárbara Oliván-Blázquez.

**Writing – review & editing:** Rafael Sánchez-Arizcuren, Esperanza García-Uceda, Bárbara Oliván-Blázquez.

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
