## [Decision Letter · Decision Letter 0]

22 Dec 2024

PONE-D-24-45286INDICATORS TO EVALUATE THE LABOUR INSERTION OF PEOPLE WITH DISABILITIES IN CONVENTIONAL COMPANIES IN SPAIN: A DELPHI STUDYPLOS ONE

Dear Dr. García-Uceda,

Thank you for submitting your manuscript to PLOS ONE. After careful consideration, we feel that it has merit but does not fully meet PLOS ONE’s publication criteria as it currently stands. Therefore, we invite you to submit a revised version of the manuscript that addresses the points raised during the review process.

A review of bibliographic references is required. The reviewers' comments are relevant and can qualify your manuscript; however, the points addressed demonstrate weaknesses that should be evaluated by the authors.

We look forward to receiving your revised manuscript.

Kind regards,

Maria Carlota Borba Brum, PhD

Academic Editor

PLOS ONE

“This work has been funded by the Department of Welfare and Family of the Government of Aragón (Spain) (OTRI Project: 2024/2009).”

4. We notice that your supplementary tables are included in the manuscript file. Please remove them and upload them with the file type 'Supporting Information'. Please ensure that each Supporting Information file has a legend listed in the manuscript after the references list.

Additional Editor Comments:

A review of bibliographic references is required.

Reviewers' comments:

Reviewer's Responses to Questions

**Comments to the Author**

1. Is the manuscript technically sound, and do the data support the conclusions?

Reviewer #1: Partly

Reviewer #2: Yes

2. Has the statistical analysis been performed appropriately and rigorously? 

Reviewer #1: Yes

Reviewer #2: Yes

3. Have the authors made all data underlying the findings in their manuscript fully available?

Reviewer #1: Yes

Reviewer #2: Yes

4. Is the manuscript presented in an intelligible fashion and written in standard English?

Reviewer #1: Yes

Reviewer #2: Yes

5. Review Comments to the Author

Reviewer #1: Interesting article in terms of its purpose and methodology, but it has limitations for publication, at least in its current version.

I list some weaknesses that should be improved, in my opinion, in order to be published in a journal of the quality of PLOS ONE

Focus of the work:

Better justification of the need for this set of indicators: For whom are they needed. Who says they are needed. Why and what they are needed for. Why for PwD and not for all workers.

Methodology:

Correct choice of technique, but: better justification for the choice of this technique (and not another). Supporting literature on the choice.

Delphi literature additional to that presented, more up-to-date. Use of references from authors with more contrasted expertise in this methodology (see some references at the end of the review)

The team conducting the research: who they are, indicators of their knowledge of the technique and the object of study.

Selection of experts: who carries it out, more explicit selection criteria.

Round 0, open-ended questions: how these questions are defined, literature supporting the content of the selected questions. Better explanation of how responses are interpreted, grouped, discarded or selected to be incorporated into the closed-ended questionnaire in phase 2. The support of a group of 5 new experts is positive, but, in my opinion, it would have been advisable to accompany it with a pilot test of the second questionnaire with another group of experts.

Third questionnaire: were they not given indicators of dispersion (standard deviation or interquartile range? were they not asked to justify their answer if they stayed out of a more or less centred position (between Q1 and Q3, for example)? were they only asked to justify their answer if they changed their opinion? In my opinion, this induces not to change one's position in order not to have to justify one's answer. It encourages a false stability, limits the exchange of reasoning and the ability of the delphi to generate real changes of opinion resulting from the possession of more information and more reflexion, leading to a higher quality and more consensual group opinion. In fact, the third round contributed practically nothing, so the second round would have been sufficient, i.e. a survey rather than a delphi.

Analysis of results

The six indicators that reflected different group opinions between the group of PWD experts and the group of contracting companies were eliminated. In my opinion these would have been the most interesting indicators to promote interaction and argumentation. In any case, I believe that the argument for incorporating or removing indicators should have been the average importance assigned to each indicator rather than the consensus reached. And if in this case the average importance was low, I think that should be highlighted.

It would be interesting to have information on the qualitative contributions made in the open question after each item in the second questionnaire and the experts who modified their opinion. Quantitative data and selection of noteworthy qualitative contributions, if applicable.

It would be interesting to accompany the work with an assessment by experts external to the research team of the quality of the development of the Delphi study, the usefulness of its results and the trustworthiness and agreement with them, based, for example, on a brief survey of the panel of participant experts.

Discussion

I believe that the discussion should focus on highlighting what the different experts consider most important for measuring the quality of the labour insertion of pwd, and whether or not they fit with what the academic literature reflects so far, accompanied by reasoned explanations for these potential mismatches, rather than making assertions that are not derived from the results of this study, such as lines 390-392 or 397-405.

In line with the suggestions made for the focus of the work, I believe that reflections on what use these indicators might be and for whom, as well as how they should be completed and by whom, should also be included in more detail in this final section.

The local character of the work should also be reflected as a limitation, and it would be advisable to explain its possibilities of generalisation to which geographical and sectoral spheres it could reach.

In sum, it is an interesting work, but in my opinion, it lacks a more focused approach and some weaknesses in its theoretical review and in the development of its fieldwork and the analysis of its results, which should be overcome before its publication.

References

• Dalkey, N., & Helmer, O. (1963). An experimental application of the Delphi method to the use of experts. Management Science, 9(3), 458-467.

• Rowe, G., & Wright, G. (1999). The Delphi technique as a forecasting tool: issues and analysis. International Journal of Forecasting, 15(4), 353-375.

• Okoli, C., & Pawlowski, S. D. (2004). The Delphi method as a research tool: an example, design considerations and applications. Information & Management, 42(1), 15-29. (

• Von der Gracht, H. A. (2012). Consensus Measurement in Delphi Studies Review and Implications for Future Quality Assurance. Technological Forecasting & Social Change, 79(8), 1525–1536 (1986)

• Belton, I., MacDonald, A., Wright, G., & Hamlin, I. (2019). Improving the practical application of the Delphi method in group-based judgment: A six-step prescription for a well-founded and defensible process. Technological Forecasting and Social Change, 147, 72-82.

• Flostrand, A., Pitt, L., & Bridson, S. (2020). The Delphi technique in forecasting–A 42-year bibliographic analysis (1975–2017). Technological Forecasting and Social Change, 150, 119773. (157)

• Andersen, P. D. (2022). Constructing Delphi statements for technology foresight. Futures & Foresight Science, e144, 1-15.

• Landeta, J., & Lertxundi, A. (2023). Quality indicators for Delphi studies. Futures & Foresight Science, e172

• Niederberger, M., Schifano, J., Deckert, S., Hirt, J., Homberg, A., Köberich, S., ... & DEWISS network. (2024). Delphi studies in social and health sciences—Recommendations for an interdisciplinary standardized reporting (DELPHISTAR). Results of a Delphi study. Plos one, 19(8), e0304651

Reviewer #2: Congrats for the good and interesting wok done but little more should be done to take it to next level.

Please work on the suggestions given, also mention why you choose those questions for phase 1, in fact the logic behind each question should be supported by literature review.

This process will enhance the reliability and validity of the questions asked and further strengthen your research and the analysis part too.

While reporting the analysis you can refer the literature for each question, and you can get better insights.

Also add points in the limitations part especially for the PWD aspect.

Final part , discussion etc is little weak strengthen it and clear state the takeaway.

Review ofmanuscript "INDICATORS TO EVALUATE THE LABOUR INSERTION OF PEOPLE WITH DISABILITIES IN CONVENTIONAL COMPANIES IN SPAIN: A DELPHI STUDY."

This manuscript focuses on developing indicators to evaluate the labor market insertion of people with disabilities (PwD) in conventional companies in Spain. Below is a critique based on various aspects of the study:

Strengths:

      1.    Timely and Relevant Topic:

The inclusion of PwD in the labor market is a globally significant issue, and the study aligns with ongoing efforts to promote inclusive workplaces.

      2.    Use of Delphi Methodology:

The iterative and consensus-driven Delphi method ensures comprehensive input from experts, lending credibility to the findings.

      3.    Comprehensive Indicator Development:

The creation of 26 indicators across four dimensions (Work Performance, Labor Management, Social and Organizational Impact, and Competency Assessment) provides a multi-faceted approach to assessing inclusivity.

      4.    Clear Ethical Standards:

Ethical approval and anonymization of data strengthen the study’s integrity and adherence to research norms.

      5.    Factor Analysis:

Statistical validation of the indicators through factor analysis increases the reliability and applicability of the framework.

Weaknesses:

      1.    Overdependence on Expert Opinion:

While the Delphi method is robust, the study heavily relies on expert opinions, which may not fully capture on-ground realities faced by PwD or employers.

      2.    Sample Size and Composition:

The sample size (48 experts) is relatively small and may limit the generalizability of the findings. Additionally, a higher representation of diverse industries could have provided broader insights.

      3.    Limited Stakeholder Inclusion:

The study focuses on HR experts, disability advocates, and company representatives but does not directly involve PwD, whose perspectives are crucial for a holistic evaluation.

      4.    Possible Bias in Indicator Selection:

Indicators were subjectively categorized and selected, which might introduce biases despite attempts to achieve consensus.

      5.    Insufficient Attention to Implementation:

The study identifies indicators but does not provide actionable strategies or tools for companies to implement and measure these effectively.

Suggestions for Improvement:

      1.    Include PwD in the Study:

Gathering feedback directly from PwD about their workplace experiences can complement expert insights and ensure the indicators address real challenges…. But may be it cant be done now so please mention this as future research

      2.    Expand the Sample:

Involving more participants from diverse industries and regions could increase the robustness and applicability of the findings.

      3.    Practical Guidance:

Offer clear guidelines or a toolkit for companies to adopt the proposed indicators and integrate them into their existing HR frameworks.

      4.    Validation in Real-World Contexts:

Pilot testing the indicators in actual workplaces could strengthen their reliability and utility.

      5.    Focus on Sustainability:

Investigate how these indicators can contribute to long-term changes in workplace inclusivity rather than short-term compliance or reputation management.

      6.    Address Technological Accessibility:

Incorporate specific indicators for the use of accessible technologies, which are pivotal for PwD inclusion in modern workplaces.

This manuscript is a solid contribution to disability inclusion research but could benefit from deeper engagement with stakeholders and real-world validation of the proposed indicators.

6. PLOS authors have the option to publish the peer review history of their article (what does this mean? ). If published, this will include your full peer review and any attached files.

**Do you want your identity to be public for this peer review?** For information about this choice, including consent withdrawal, please see our Privacy Policy .

Reviewer #1: No

Reviewer #2: No

---

## [Author Response · Author response to Decision Letter 1]

14 Feb 2025

Review of manuscript "INDICATORS TO EVALUATE THE LABOUR INSERTION OF PEOPLE WITH DISABILITIES IN CONVENTIONAL COMPANIES IN SPAIN: A DELPHI STUDY." 37

First of all, we appreciate the effort and time invested in this manuscript. These are very valuable comments that we are sure, they enhance its quality.

We will proceed to respond point by point to the comments from the editor and reviewers.

EDITOR´S COMMENTS

A review of bibliographic references is required.

Thank you for your comments. A review of the bibliographic references has been done, and some references has been added:

[3] World Health Organization (WHO). World report on disability. 2011. Available from: https://www.who.int/es/publications/i/item/9789241564182

[4] Fullana Noell, J. F., Díaz, M. P., & Suñé, M. V. (2003). [Research on the labor integration processes of people with disabilities in ordinary settings. A qualitative case study]. Revista de Investigación Educativa. 2003; 21(2): 305-321. Available from: [http://riberdis.cedid.es/bitstream/handle/11181/3369/Investigacion_sobre_procesos_integracion_laboral_personas_con_discapacidad.pdf]

[5] State Public Employment Service (SEPE). [Report on the labor market for people with disabilities. Spain. Data 2023.]. 2024. Available from: [https://sid-inico.usal.es/wp-content/uploads/2024/04/Informe-del-Mercado-de-Trabajo-2023.pdf]

[7] Royal Board on Disability (RPsD), Ministry of Labor and Social Economy, CERMI, & ONCE Foundation. [White Paper on Employment and Disability]. 2023. Available from: [https://www.rpdiscapacidad.gob.es/estudios-publicaciones/libro_blanco_empleo_discapacidad.htm]

[9] Ishikawa, K., & Loftus, J. H. Introduction to quality control. Chapman & Hall. 1991.

[10] Peiró, JM. [Introduction to work psychology]. Madrid: Ediciones CEF. 2013

[16] Villa N. [Employment situation of people with disabilities in Spain]. Rev. complut. Educ. 2003;14(2):393-424

[19] Fundación ONCE & Fundación Manpower. [Study on the perspectives of the business world with respect to the hiring of people with disabilities]. Madrid: Fundación ONCE. 2008. Available from: [https://consaludmental.org/publicaciones/Perspectivasempresarialcontratacionpersonasdiscapacidad.pdf]

[21] Mira-Aladrén M, Yagüe-Sanjuán N, Latorre-Dena F. [Disability, inclusion and public employment, is it possible to improve their relationship? Reflections from a pilot experience at the University of Zaragoza]. Acciones e Investigación Sociales. 2024; (45). Available from: [https://papiro.unizar.es/ojs/index.php/ais/article/view/9024]

[26] Fundación ONCE. [Incentives for hiring people with disabilities]. Fundación ONCE. 2023. Available from: https://www.portalento.es/IncentivosContratacionPCD.pdf]

[27] Sánchez, A., Fernández, M.P., Pérez Nieto, M.A., Carpintero, E., Pérez Fernández, F. y Mampaso, J. [Study on the social and labor reality of people with disabilities in the Community of Madrid: analysis of the existing legislative measures of insertion and degree of compliance with them by companies]. Madrid: UCJC. 2008.

[29] Niederberger, M., Schifano, J., Deckert, S., Hirt, J., Homberg, A., Köberich, S., ... & DEWISS network. Delphi studies in social and health sciences—Recommendations for an interdisciplinary standardized reporting (DELPHISTAR). Results of a Delphi study. Plos one. 2024; 19(8): e0304651. DOI: 10.1371/journal.pone.0304651

[30] Von der Gracht, H. A. Consensus Measurement in Delphi Studies Review and Implications for Future Quality Assurance. Technological Forecasting & Social Change. 2012; 79(8): 1525–1536 (1986). DOI: 10.1016/j.techfore.2012.04.013

[31] Carl J, Mazzoli E, Mouton A, Sum RK-W, Singh A, Niederberger M, et al. (2024) Development of a Global Physical Literacy (GloPL) Action Framework: Study protocol for a consensus process. PLoS ONE. 2024; 19(8): e0307000. DOI: 10.1371/journal.pone.0307000

[32] Okoli, C., & Pawlowski, S. D. The Delphi method as a research tool: an example, design considerations and applications. Information & Management. 2004; 42(1),:15-29. DOI: 10.1016/j.im.2003.11.002

[33] Flostrand, A., Pitt, L., & Bridson, S. The Delphi technique in forecasting–A 42-year bibliographic analysis (1975–2017). Technological Forecasting and Social Change. 2020; 150: 119773. (157). DOI: 10.1016/j.techfore.2019.119773

[34] Dalkey, N., & Helmer, O. (1963). An experimental application of the Delphi method to the use of experts. Management science. 1963; 9(3): 458-467.

[36] Rowe, G., y Wright, G. The Delphi technique as a forecasting tool: issues and analysis. International Journal of Forecasting. 1999; 15(4): 353-375.

[37] Belton, I., MacDonald, A., Wright, G., & Hamlin, I. Improving the practical application of the Delphi method in group-based judgment: A six-step prescription for a well-founded and defensible process. Technological Forecasting and Social Change. 2019; 147: 72-82. DOI: 10.1016/j.techfore.2019.07.002

[38] García-Uceda, E. y Murillo-Luna, JL [Application of the Delphi method for the analysis of the determinants of Social Entrepreneurship.]. Ponencia. XXVII Jornadas Hispano_Lusas de Gestión Científica. Localización y dinámicas competitivas en un entorno global. Libro de ACTAS/ Resúmenes. 2017. IBSN: 978-84-16724-37-6; 1 – 4 febrero. Benidorm (Alicante).

[40] Martínez E. The Delphi Technique as a stakeholder consultation strategy for programme evaluation. Revista de Investigación Educativa. 2003: 21(2): 449-163.

[41] García, L., Fernández, S. Procedimiento de aplicación del trabajo creativo en grupo de expertos. Energética, XXIX (2). 2008; 46-50.

[42] Devaney, L., & Henchion, M. (2018). Who is a Delphi ‘expert’? Reflections on a bioeconomy expert selection procedure from Ireland. Futures; 2018; 99: 45-55. DOI: 10.1016/J.FUTURES.2018.03.017.

[45] Boulkedid, R., Abdoul, H., Loustau, M., Sibony, O., & Alberti, C. (2011). Using and Reporting the Delphi Method for Selecting Healthcare Quality Indicators: A Systematic Review. PLoS ONE. 2011;6(6):e20476. doi: 10.1371/journal.pone.00204766.

[50] Landeta, J., y Lertxundi, A. Quality indicators for Delphi studies. Futures & Foresight Science. 2024; 6(1). e172. DOI: 10.1002/ffo2.172

[51] Niederberger, M., & Spranger, J. Delphi Technique in Health Sciences: A Map. Frontiers in public health. 2020; 8: 457. DOI: 10.3389/fpubh.2020.00457.

[52] Skulmoski, G. J., Hartman, F. T., & Krahn, J. The Delphi method for graduate research. Journal of Information Technology Education: Research. 2007; 6(1), 1-21. DOI: 10.28945/199

[53] Niederberger, M., Köberich, S., & DeWiss Network. Coming to consensus: the Delphi technique. European Journal of Cardiovascular Nursing. 2021; 20: 692–695. DOI:10.1093/eurjcn/zvab059

[56] Valls, M. J., Vila, M & Pallisera, M. [The insertion of people with disabilities in ordinary work. The role of the family]. Revista de educación. 2004, 334(1): 97-117.

[61] Pico Barrionuevo, F. P., y Torres, S. S. [Best corporate social responsibility practices in the inclusion of people with disabilities. A case study of companies in Ambato, Ecuador]. Retos. 2017; 7(14). DOI: 10.17163/ret.n14.2017.10

[62] Pupiales, B. E., y Andrade, L. C. [The labor inclusion of people with disabilities: An ethnographic study in five autonomous communities in Spain.]. Archivos de Medicina (Manizales). 2016; 16(2): 279-289. DOI: 10.30554/ARCHMED.16.2.1720.2016

[66] Ortiz-Marcos, J. M., Invernón-Gómez, A. I., Medina-García, M., & Higueras-Rodríguez, L. [Labor inclusion of people with disabilities: Legislative challenges and practical solutions.]. Revista Española de Orientación y Psicopedagogía. 2024; 35(3): 105-122. DOI: reop.vol.35.num.3.2024.40820

[70] Kuznetsova, Y., Yalcin, B. Inclusion of persons with disabilities in mainstream employment: is it really all about the money? A case study of four large companies in Norway and Sweden. Disability & Society. 2017; 32(2): 233–253. DOI:10.1080/09687599.2017.1281794

[71] Shaw, J., Wickenden, M., Thompson, S., & Mader, P. Achieving disability inclusive employment – Are the current approaches deep enough? Journal of International Development. 2022; 34(5): 2022, 942–963. DOI: 10.1002/jid.3692

[72] Darcy, S., Taylor, T., & Green, J. “But I can do the job”: examining disability employment practice through human rights complaint cases. Disability & Society. 2016; 31(9): 1242–1274. DOI: 10.1080/09687599.2016.1256807

[73] Betancourt, D. C., Echeverri, M. P. G., & Rubio, L. A. Working Inclusion: A Pending Commitment of Organizations towards People with Disabilities. Siglo Cero. 2024; 55(1): 49–65. DOI: 10.14201/scero.31335

[74] Pallisera, M., Vila, M., Valls, M. J., Rius, M., Fullama, J., Jiménez, P., Cardona, M., & Lobato, J. [The labor integration of people with disabilities in the ordinary company in Spain: a research approach.]. Revista Española sobre Discapacidad Intelectual. 2003; 34: 5–18.

[75] Shore, C., Wright, S., & Però, D. (Eds.). Policy worlds: Anthropology and the analysis of contemporary power (Vol. 14). Berghahn Books. 2011.

[76] Robinson, G. and Dechant, K. (1997) Building a Business Case for Diversity. Academy of Management Executive, 1997; 11: 21-31. DOI: 10.5465/AME.1997.9709231661

[77] Herring, C. Does diversity pay?: Race, gender, and the business case for diversity. American sociological review. 2009; 74(2): 208-224. DOI: 10.1177/000312240907400203

[78] Stone, D. L., & Colella, A. A model of factors affecting the treatment of disabled individuals in organizations. Academy of management review. 1996, 21(2), 352-401.

[79] Booth, A., & Leigh, A. Do employers discriminate by gender? A field experiment in female-dominated occupations. Economics Letters. 2010; 107(2): 236-238. DOI: 10.1016/j.econlet.2010.01.034

[80] Lindsay, S., Osten, V., Rezai, M., & Bui, S. Disclosure and workplace accommodations for people with autism: a systematic review. Disability and rehabilitation. 2021; 43(5): 597–610. DOI: 10.1080/09638288.2019.1635658

[81] Chen A, O’Neill J, Phillips KG, Houtenville AJ, Katz E. Relationship of business practices and characteristics to supervisors’ perceived effectiveness of disability recruitment. Journal of Vocational Rehabilitation. 2023;59(3):301-310. DOI:10.3233/JVR-230047

[82] Rezai, M., Kolne, K., Bui, S. et al. Measures of Workplace Inclusion: A Systematic Review Using the COSMIN Methodology. J Occup Rehabil. 2020; 30: 420–454. DOI: 10.1007/s10926-020-09872-4

[83] Adjo, J., Maybank, A., & Prakash, V. Building inclusive work environments. Pediatrics, 2021;148(Supplement 2). DOI: 10.1542/peds.2021-051440E

[84] Watkins, M. W. Exploratory factor analysis: A guide to best practice. Journal of black psychology. 2018; 44(3): 219-246. DOI: 10.1177/0095798418771807

In addition, the updating of contents has led to the elimination of some references, such as the following:

- International Labour Organization (ILO). World report on disability. 2020. Available from: [https://www.ilo.org/global/publications/books/WCMS_734617/lang--en/index.htm]

- Palomo R. [Factors influencing the employment rate of people with disabilities]. Informe Anual de Empleo, 2012, 14, 123-135.

- National Statistical Institute (INE). [Disabled People's Labour Market Report]. 2023. Available from: [https://www.ine.es/prensa/epd_2022.pdf]

- Taborda Morales, Y. L. T. [Business productivity and inclusion of people with disabilities: challenges and opportunities]. Revista Perspectiva Empresarial. 2023. 10(1), 3-5. DOI: 10.16967/23898186.808

- Barragán R. The inclusion of people with disabilities in the labour market: challenges and opportunities. Journal of Disability Policy Studies. 2017, 27(3): 131-140. DOI: 10.1177/1044207316656325

A reorganization of the citation in the manuscript has been performed.

REVIEWER 1

Interesting article in terms of its purpose and methodology, but it has limitations for publication, at least in its current version.

I list some weaknesses that should be improved, in my opinion, in order to be published in a journal of the quality of PLOS ONE

Thank you very much for your comments. They have been very useful to improve the quality of the manuscript.

Better justification of the need for this set of indicators: For whom are they needed. Who says they are needed. Why and what they are needed for. Why for PwD and not for all workers.

The rationale for the study has been expanded in the revised version to address in greater detail the specific challenges faced by people with disabilities (PwD) in their access to and development in the labor market. These challenges include structural, cultural and attitudinal barriers that limit their employment opportunities and hinder their full integration into mainstream work environments. Labor inclusion not only implies access to employment, but also the creation of conditions for PwD to thrive and develop their potential.

The lack of comprehensive tools to assess the labor inclusion of PwDs beyond hiring rates has also been highlighted. Although these rates are important indicators, they do not capture key aspects such as organizational climate, accessibility and the impact of adaptations in the work environment. This shortcoming hinders the implementation of comprehensive strategies to promote inclusive work environments.

Finally, references to national and international policy frameworks that underline the relevance of inclusive practices adapted to the needs of PwD have been incorporated. Documents such as the UN Convention on the Rights of Persons with Disabilities, the European Disability Strategy 2021-2030 and the White Paper on Employment and Disability (Spain) provide clear guidelines for the promotion of labor inclusion provide clear guidelines for the promotion of labor inclusion. These frameworks emphasize the need to go beyond general diversity measures by proposing specific approaches to ensure equal participation of PwDs in the labor market.

The following has been added:

Lines 45-54: However, measuring success in workplace inclusion remains a challenge, due to the lack of specific tools that assess both individual adaptations and organizational practices. This issue not only responds to an ethical and human rights imperative [1], but also offers specific benefits for businesses and the economy in general [2]. According to the International Report on Disability, around one billion people, or 15% of the world's population, live with some form of disability and many of them face significant barriers to access formal employment [3]. In this context, having specific indicators would facilitate the objective evaluation of inclusive policies and practices, allowing companies to measure their impact and improve their strategies.

Lines 59-60: In Spain, people with disabilities represent 6.33% of the working-age population, but their activity rate is significantly lower, reaching 35.3% in 2023 [5]. This gap highlights the need for instruments to identify not only the existing barriers, but also the actions required to overcome them and evaluate their effectiveness.

Lines 65-67: In Spain, the White Paper on Employment and Disability proposes a legislative and public policy framework for the employment and right to work of people with disabilities [7].

Lines 71-74: The development of specific indicators that go beyond hiring rates and are adapted to the characteristics of PwD is crucial to evaluate the success of their inclusion in the regular company as well as to encourage the implementation of inclusive practices [5,7].

Methodology:

Correct choice of technique, but: better justification for the choice of this technique (and not another). Supporting literature on the choice.

Additional literature has been incorporated to strengthen the methodological justification, including references, which underline the Delphi method’s appropriateness for addressing complex, multidisciplinary problems with limited empirical evidence.

Delphi literature additional to that presented, more up-to-date. Use of references from authors with more contrasted expertise in this methodology (see some references at the end of the review)

Some references have been cited and included in the text.

[29] Niederberger, M., Schifano, J., Deckert, S., Hirt, J., Homberg, A., Köberich, S., ... & DEWISS network. Delphi studies in social and health sciences—Recommendations for an interdis

---

## [Decision Letter · Decision Letter 1]

28 Feb 2025

PONE-D-24-45286R1INDICATORS TO EVALUATE THE LABOR INSERTION OF PEOPLE WITH DISABILITIES IN CONVENTIONAL COMPANIES IN SPAIN: A DELPHI STUDY.PLOS ONE

Dear Dr. García-Uceda, 

Thank you for submitting your manuscript to PLOS ONE. After careful consideration, we feel that it has merit but does not fully meet PLOS ONE’s publication criteria as it currently stands. Therefore, we invite you to submit a revised version of the manuscript that addresses the points raised during the review process.

We look forward to receiving your revised manuscript.

Kind regards,

Maria Carlota Borba Brum, PhD

Academic Editor

PLOS ONE

**Additional Editor Comments:**

This manuscript is a contribution to disability inclusion research but the methodological shortcomings partially support the authors' conclusions, as pointed out by the reviewers, although it has presented significant improvements. These limitations require actions by t.he research team so that its publication can be evaluated

Reviewers' comments:

Reviewer's Responses to Questions

**Comments to the Author**

1. If the authors have adequately addressed your comments raised in a previous round of review and you feel that this manuscript is now acceptable for publication, you may indicate that here to bypass the “Comments to the Author” section, enter your conflict of interest statement in the “Confidential to Editor” section, and submit your "Accept" recommendation.

Reviewer #1: (No Response)

2. Is the manuscript technically sound, and do the data support the conclusions?

Reviewer #1: Partly

3. Has the statistical analysis been performed appropriately and rigorously? 

Reviewer #1: Yes

4. Have the authors made all data underlying the findings in their manuscript fully available?

Reviewer #1: Yes

5. Is the manuscript presented in an intelligible fashion and written in standard English?

Reviewer #1: Yes

6. Review Comments to the Author

Reviewer #1: The authors have made a great effort to complete the suggestions made by the reviewers. Congratulations on that part.

As far as my comments are concerned, I reaffirm that I find the article interesting in terms of its purpose and methodology. But I continue to think that it still has weaknesses that should be remedied, as far as possible, before it is published in PLOS.

The authors have completed and enriched the article considerably. I think it is much improved, but it still has a strong problem stemming from the design of the Delphi application: in my opinion, the main weakness is that it has focused too much on reaching a high degree of consensus. However, consensus can be a purpose of the Delphi, but only when it is real, i.e. when it is achieved by changes in initial opinions resulting from further reflection and internalisation of the rationale provided by the other experts. The rationales provided by the experts have been gathered by the research team, which has helped them to better interpret the indicators, which is good, but they have not been transferred to the rest of the experts. There has been no interaction of content and opinions among the participating experts, selected and mediated by the research team.

When the group average and the expert's previous estimate is the only feedback received, and when the change of opinion has the additional penalty of having to justify it, the consensus reached may not be real. Even more so when the items with the most dissent have been eliminated for that reason. That is why I maintain my view that the third round has contributed almost nothing, or may even have contributed a false sense of consensus, which is worse than nothing.

This fact should be reflected in the limitations of the paper, although I think this would be necessary but not sufficient. The authors have circumvented many of the more profound suggestions for improvement they have received by expanding the limitations section or proposing their incorporation in future work. Understandably, but I think something could be done already in this one, in order to be published. I understand that it is not feasible to repeat the fieldwork, because it has already been done, but it is possible, at least, to obtain validation from the participating experts on the degree to which the method has been effective in obtaining input from the expert panel members, the degree to which the feedback received has helped them to improve their confidence in the answers given, their confidence in the technique or methodology used, the quality and clarity of the presentation of the questions, their satisfaction with participation in the study, and, above all, the usefulness of the results achieved and their confidence in them.

It would involve contacting the experts again and asking them to respond to this short survey. In this way, if the results were favourable, especially in terms of confidence in the results and their usefulness, the main contribution of the article, the proposal of indicators, would be endorsed by the experts who have contributed to its generation, and the results would gain in credibility, with methodological shortcomings taking second place. If it is not clearly endorsed by them, then it should not be published and the research should be conducted again, incorporating methodological improvements.

Other specific areas for improvement:

Reference 38, from the authors themselves: Summary of a conference proceedings. It does not allow consultation and should not be included among the relevant methodological references.

Text introduced in lines 226-273: Unnecessarily repeats some ideas in paragraphs 235-241 with those in paragraph 256-259.

Text introduced on lines 670-674 is repeated in 676-679.

I encourage the authors to undertake the proposed improvements, as the resulting contribution may be worth the effort. I am still grateful to an anonymous reviewer who many years ago suggested to me to do something similar. Despite the initial discouragement and the effort it took, the work gained credibility, and I learned a lot from it.

7. PLOS authors have the option to publish the peer review history of their article (what does this mean? ). If published, this will include your full peer review and any attached files.

**Do you want your identity to be public for this peer review?** For information about this choice, including consent withdrawal, please see our Privacy Policy .

Reviewer #1: No

---

## [Author Response · Author response to Decision Letter 2]

23 Mar 2025

Dear PhD. Borba Brum,

First of all, we appreciate the effort and time invested in this manuscript. These are very valuable comments that we are sure, they enhance its quality.

We will proceed to respond point by point to the comments from the reviewers.

1. On the lack of interaction among experts in the Delphi process

We understand your concern regarding the lack of interaction among the experts and the possibility that the consensus reached did not reflect a genuine change in opinions. To address this concern, we conducted additional validation through a follow-up survey of the experts who participated in the study. This survey assessed their perception of the clarity of the process, the usefulness of the feedback received, the influence of the group mean, and the need for the third round.

The following has been added:

Lines 357-366: “To complement and validate the results obtained in the main study, a follow-up survey was conducted among the experts who participated in the Delphi process. The objective of this survey was to evaluate the clarity of the process, the usefulness of the feedback received in each round, the impact of the justification of changes, the perception of the need for the third round and the applicability of the indicators developed. The survey consisted of 11 Likert-scale questions, where participants rated each item on a scale from 1 (extremely unclear, not useful, strongly disagree, etc.) to 10 (extremely clear, highly useful, strongly agree, etc.). Additionally, an open-ended section allowed experts to provide qualitative feedback. The survey was answered by 21 experts and the results were analyzed using descriptive statistics and qualitative analysis of open comments.”

The results indicated that the Delphi process was considered clear (M=8.14, SD=1.06) and that the feedback received helped to value the responses (M=7.95, SD=1.43). However, the third round was not perceived as essential (M=3.95, SD=0.97), reinforcing the need to review its impact in future research.

The following has been added:

Lines 559-571: “The results of the validation survey confirmed that most experts considered the Delphi process to be clear (M=8.14, SD=1.06) and that the feedback received was useful for reflecting on their answers (M=7.95, SD=1.43). However, the third round was not perceived as essential (M=3.95, SD=0.97), reinforcing the need to discuss its impact on consensus generation. Regarding the applicability of the indicators in labor inclusion, the experts considered them useful (M=7.86, SD=1.19) and applicable to design labor inclusion strategies (M=8.19, SD=1.40).

Table 7: Results of the validation survey

Mean Median SD IQR Min Max

Regarding the Delphi process used in the study, how did you find it? 8.14 8 1.062 1.00 6 10

To what extent did the feedback received in each round help you to evaluate your answers? 7.95 8 1.431 2.00 5 10

In the second round, do you think the information provided (group average and your previous response) was sufficient to reflect on your answers? 8.33 9 1.155 1.00 6 10

Do you consider that the justification for the changes in the answers may have influenced your decision to modify or maintain your answers? 7.19 7 1.806 3.00 3 10

To what extent do you believe that the consensus reached reflects real and substantiated expert opinions? 8.14 9 1.352 2.00 6 10

Do you think the third round was necessary to consolidate the consensus or improve the quality of the responses? 3.95 4 0.973 2.00 2 5

Do you consider that the indicators resulting from the study are applicable to the evaluation of labor inclusion in conventional companies? 7.86 8 1.195 2.00 6 10

Do you believe that the results obtained in this study are of practical use in the design of labor inclusion strategies? 8.19 8 1.401 2.00 5 10

Would you recommend conducting future studies that include validation in real work environments before applying these indicators on a large scale? 8.10 9 1.670 2.00 4 10

In addition to quantitative responses, some experts provided additional comments on the Delphi process. While several participants highlighted the value of the method for structuring the indicator assessment, others suggested that more interaction among the experts could have enriched the discussion and consensus process.”

2. On the possibility of a false sense of consensus

Although the third round was not considered fundamental by some experts, the survey confirmed that the consensus reached reflected informed opinions (M=8.14, SD=1.35). To mitigate this limitation, we have included in the discussion and limitations of the study an explicit acknowledgement of this observation, as well as the need to include more dynamic interactions in future applications of the Delphi method.

The following has been added:

Lines: 662-671: “Although the results of the follow-up survey indicated that the third round of the Delphi process was not perceived as essential by some experts, they also validated the usefulness of the indicators generated. This suggests that, although methodological design can be improved in future studies, the results obtained are robust and relevant. Validation in real workplace settings is recognized as a crucial next step to strengthen the applicability of the indicators.

Qualitative comments from the experts reflect mixed perceptions of the Delphi process. While several participants emphasized the usefulness of the method in identifying key indicators, some suggested that more interaction among experts might have supported a more dynamic consensus. In future research, it would be advisable to promote more interaction among experts.”

Lines: 705-709: “While the Delphi method used in this study effectively structured expert input, it did not facilitate direct interaction among participants, which may have limited opportunities for more in-depth discussion. Additionally, some experts perceived the third round as less essential for refining consensus. However, a follow-up validation survey reinforced the relevance of the indicators developed, confirming their perceived usefulness and applicability in assessing labor inclusión.”

3. Reference 38, from the authors themselves: Summary of a conference proceedings. It does not allow consultation and should not be included among the relevant methodological references.

We appreciate your comment on Reference 38, which referred to an abstract of a conference proceedings presented by one of the authors. We understand that this type of reference is not easily accessible or verifiable, so we have proceeded to remove it from the manuscript.

4. Text introduced in lines 226-273: Unnecessarily repeats some ideas in paragraphs 235-241 with those in paragraph 256-259.

We appreciate your comment on the repetition of ideas in lines 226-273, 235-241, and 256-259. In response to your comment, we have carefully revised these sections and removed redundant paragraphs to improve the flow and clarity of the manuscript.

The following has been changed:

Lines: 225-227: “Phase 1 applied expert judgment to identify key indicators of inclusive employability. The first questionnaire was designed with open-ended questions to gather preliminary insights on labor inclusion for people with disabilities.”

Lines 233-238: “The development of these questions followed a rigorous process, grounded in a comprehensive literature review on labor inclusion and human resource management. Key sources such as Peiró & Prieto [11], Schur et al. [46], and Lindsay et al. [47], guided the formulation of questions addressing critical aspects of recruitment and selection. For example, one question asked: “What criteria or aspects are important when evaluating the recruitment and selection process of people with disabilities in a conventional company? “

Lines 251-253: “This review focused on areas such as recruitment practices, workplace culture, and specific employment barriers, which provided the basic framework for developing the open-ended questions.”

5. Text introduced on lines 670-674 is repeated in 676-679.

The following has been deleted:

“Although the study design included a carefully selected panel of experts and iterative validation of the indicators during the early phases, the incorporation of additional review by external experts at this stage could have enriched the external validity and applicability of the results.”

We sincerely appreciate your support and your motivation to implement the proposed improvements. Your experience encourages us to take on this challenge in order to strengthen the credibility and impact of our work.

We are convinced that these changes will not only benefit the quality of the article, but have also allowed us to deepen our learning about the application of the Delphi method and its validation. We thank you again for your valuable guidance and for taking the time to provide us with detailed and constructive comments.

Kind regards,

Esperanza García-Uceda, PhD

---

## [Decision Letter · Decision Letter 2]

28 Mar 2025

INDICATORS TO EVALUATE THE LABOR INSERTION OF PEOPLE WITH DISABILITIES IN CONVENTIONAL COMPANIES IN SPAIN: A DELPHI STUDY.

PONE-D-24-45286R2

Dear Dr. García-Uceda,

We’re pleased to inform you that your manuscript has been judged scientifically suitable for publication and will be formally accepted for publication once it meets all outstanding technical requirements.

Kind regards,

Maria Carlota Borba Brum, PhD

Academic Editor

PLOS ONE

Additional Editor Comments (optional):

Reviewers' comments:

Reviewer's Responses to Questions

**Comments to the Author**

1. If the authors have adequately addressed your comments raised in a previous round of review and you feel that this manuscript is now acceptable for publication, you may indicate that here to bypass the “Comments to the Author” section, enter your conflict of interest statement in the “Confidential to Editor” section, and submit your "Accept" recommendation.

Reviewer #1: All comments have been addressed

2. Is the manuscript technically sound, and do the data support the conclusions?

Reviewer #1: Yes

3. Has the statistical analysis been performed appropriately and rigorously? 

Reviewer #1: Yes

4. Have the authors made all data underlying the findings in their manuscript fully available?

Reviewer #1: Yes

5. Is the manuscript presented in an intelligible fashion and written in standard English?

Reviewer #1: Yes

6. Review Comments to the Author

Reviewer #1: All comments and suggestions have been incorporated, as far as possible, into the article.

I congratulate the authors for their efforts and trust that their work will achieve the social and academic usefulness it is intended to achieve.

7. PLOS authors have the option to publish the peer review history of their article (what does this mean? ). If published, this will include your full peer review and any attached files.

**Do you want your identity to be public for this peer review?** For information about this choice, including consent withdrawal, please see our Privacy Policy .

Reviewer #1: **Yes: ** Jon Landeta

---

## [Editor Report · Acceptance letter]

PONE-D-24-45286R2

PLOS ONE

Dear Dr. García-Uceda,

I'm pleased to inform you that your manuscript has been deemed suitable for publication in PLOS ONE. Congratulations! Your manuscript is now being handed over to our production team.

Kind regards,

on behalf of

Dr. Maria Carlota Borba Brum

Academic Editor

PLOS ONE